# A cattle graph genome incorporating global breed diversity

A. Talenti [1✉], J. Powell [1], J. D. Hemmink [1,2,3,4], E. A. J. Cook [2,4], D. Wragg[1,3], S. Jayaraman [1], E. Paxton[1], C. Ezeasor[5], E. T. Obishakin [6,7], E. R. Agusi[6,7], A. Tijjani [8,9], W. Amanyire[10], D. Muhanguzi[10], K. Marshall[2,4], A. Fisch [11], B. R. Ferreira [11], A. Qasim [12], U. Chaudhry[1], P. Wiener[1], P. Toye[2,4], L. J. Morrison[1,3], T. Connelley[1,3] & J. G. D. Prendergast [1,3✉]

Despite only 8% of cattle being found in Europe, European breeds dominate current genetic resources. This adversely impacts cattle research in other important global cattle breeds, especially those from Africa for which genomic resources are particularly limited, despite their disproportionate importance to the continent's economies. To mitigate this issue, we have generated assemblies of African breeds, which have been integrated with genomic data for 294 diverse cattle into a graph genome that incorporates global cattle diversity. We illustrate how this more representative reference assembly contains an extra 116.1 Mb (4.2%) of sequence absent from the current Hereford sequence and consequently inaccessible to current studies. We further demonstrate how using this graph genome increases read mapping rates, reduces allelic biases and improves the agreement of structural variant calling with independent optical mapping data. Consequently, we present an improved, more representative, reference assembly that will improve global cattle research.

[1] The Roslin Institute, Royal (Dick) School of Veterinary Studies, University of Edinburgh, Easter Bush Campus, Midlothian EH25 9RG, UK. [2] The International Livestock Research Institute, PO Box 30709 Nairobi, Kenya. [3] Centre for Tropical Livestock Genetics and Health, Easter Bush, Midlothian EH25 9RG, UK. [4] Centre for Tropical Livestock Genetics and Health, ILRI Kenya, Nairobi 30709-00100, Kenya. [5] Department of Veterinary Pathology and Microbiology, University of Nigeria, Nsukka, Enugu State, Nigeria. [6] Biotechnology Division, National Veterinary Research Institute, Vom, Plateau State, Nigeria. [7] Biomedical Research Centre, Ghent University Global Campus, Songdo, Incheon, South Korea. [8] International Livestock Research Institute (ILRI) PO, 5689 Addis Ababa, Ethiopia. [9] Centre for Tropical Livestock Genetics and Health (CTLGH), ILRI Ethiopia, PO Box 5689 Addis Ababa, Ethiopia. [10] School of Biosecurity, Biotechnology and Laboratory Sciences (SBLS), College of Veterinary Medicine, Animal Resources and Biosecurity, Makerere University, P.O Box 7062 Kampala, Uganda. [11] Ribeirão Preto College of Nursing, University of Sao Paulo, Ribeirão Preto, SP, Brazil. [12] Faculty of Veterinary and Animal Sciences, Gomal University, Dera Ismail Khan, Pakistan. ✉email: Andrea.Talenti@ed.ac.uk; James.Prendergast@roslin.ed.ac.uk

Cattle are one of the most populous farmed animals worldwide, with their global population of almost one billion seconds only to chickens[1]. Due to their use as draft animals and their ability to convert low-quality forage into energy-dense muscle and milk, they provide a significant source of nutrition and livelihood to over 6 billion people. Since their domestication almost 10,000 years ago, hundreds of distinct cattle breeds have been established, displaying a diverse range of heritable phenotypes, from differences in production phenotypes such as milk yield, to environmental adaptation, disease tolerance and altered physical characteristics such as horn shape and skin pigmentation[2,3].

This phenotypic diversity between cattle breeds is mirrored by substantial genetic diversity, but this is poorly reflected by current reference resources. The primary reference genome is derived from a single European Hereford cow[4] and projects such as the 1000 bulls genomes project are heavily skewed towards European-derived breeds (*Bos taurus taurus*) due to a number of factors such as geographic distribution and sample accessibility[5]. Although European breeds largely all originate from the same domestication event that occurred in the Middle East, at least one further domestication event occurred in South Asia giving rise to the humped indicine breeds (*Bos taurus indicus*)[6]. These two *Bos* lineages have been estimated to have last had a common ancestor over 210,000 years ago[7] meaning the current Hereford reference genome particularly poorly represents the indicus sub-species.

As well as this primary split, it has been suggested that introgression with further Auroch populations has occurred in Africa, with the adaptation of certain African cattle breeds to local diseases potentially the result of this historical introgression[6]. In Africa alone there are over 150 indigenous cattle breeds, and almost 350 million head of cattle making up 23% of the global cattle population[1]. This compares to only 8% of cattle being located in Europe. Africa's unique history, with multiple waves of migration of both *Bos indicus* and *Bos taurus* cattle into the continent, along with its variety of environments, pathogens and cultures have led to unusually high levels of diversity among the cattle in the region. However, this diversity is not reflected in the genomic resources currently available.

The reliance of cattle research on the European Hereford reference genome has two main limitations. First, because it represents one consensus haplotype of a single animal, large sections of the cattle pan-genome are missing from this reference sequence. This is exemplified by a recent human study that identified almost 300 million bases of DNA among African individuals that were missing from the human reference genome[8]. This DNA sequence, equivalent to 10% of the human pan-genome, is consequently inaccessible to studies reliant upon the current human reference genome. The second major limitation, common to all linear reference genomes, is that even where they contain the region being studied, downstream analyses are biased towards the alleles and haplotypes present in the reference sequence[9,10]. The emerging field of graph genomes aims to address these issues by incorporating genetic variation and polymorphic haplotypes as alternative paths within a single graph representation of the genome. This has the advantage that reads which do not directly match a linear reference may still perfectly match a route through the graph, increasing the accuracy of read alignment. Several recent studies have highlighted how the use of such genome graphs can increase read mapping and variant calling accuracy, reduce mapping biases[11,12], identify ChIP-seq peaks not identified using linear genomes[13,14], and better characterise transcription factor motifs[15]. However, there are currently few high-quality graph genomes available. In livestock, the use of graph genomes has so far been restricted to studies simply incorporating variants from short-read sequencing data into the

Hereford reference[16,17] or to only large (>100 bp) differences between non-Hereford assemblies with the ARS-UCD1.2 reference genome[18]. Although not able to capture wider cattle diversity, these studies illustrated that the variant calls using the graph genome were more consistent between sire-son pairs than those obtained using the linear Hereford reference, with the current standard variant calling algorithms GATK HaplotypeCaller[19] and FreeBayes[20]. Graph genomes consequently have the potential to improve the detection of genetic variants, including those potentially driving important phenotypic differences between populations and breeds. However, the construction of high-quality graph genomes is dependent upon the availability of representative reference sequences, a resource that has been largely lacking for non-European cattle.

In this study, we address the current lack of reference genomes for African cattle breeds by generating assemblies for the N'Dama and Ankole breeds. These breeds display tolerance to two of Africa's most important livestock diseases; African Animal Trypanosomiasis (AAT), a disease that costs African livestock farmers billions of dollars a year[21], and East Coast fever (caused by *Theileria parva*), which causes an annual economic burden of ~$600 million[22]. We combine these genomes with three public reference assemblies representing Hereford, Angus and Brahman cattle, along with genetic variation data for 294 animals representative of global cattle breeds[23], to provide a high-quality cattle graph genome spanning global breed diversity. We go on to show how this novel, more representative, cattle graph genome can substantially improve omics studies across global cattle breeds relative to the standard primary Hereford reference.

## Results

**Generating African genome assemblies.** Global cattle breeds display high levels of genetic diversity (Fig. 1). Whereas European breeds represent only a small fraction of this diversity, African breeds display a broad spectrum of indicine to taurine variation. As the currently published Hereford[4], Brahman[24] and Angus[24] genomes poorly represent global diversity, and in particular that found in Africa, we generated two new assemblies for the West African Taurine N'Dama and East African Sanga Ankole (an ancient stabilized cross between indicine and taurine breeds). We sequenced the genomes of N'Dama and Ankole bulls at an approximate coverage of 40X Pac Bio long-read data for the assembly process and 70X of Illumina paired end reads for the genome polishing. The N'Dama contigs were scaffolded using the previously published cattle genomes, whereas the Ankole was scaffold using 100X of novel monocyte-derived bionano data. The genomes consisted of 1210 and 7581 sequences with scaffold N50s of 104.8 Mb and 84.5 Mb for the N'Dama and Ankole genomes, respectively. The final contig N50s were 10.7 Mb and 18.6 Mb for the N'Dama and the Ankole respectively, with total genome lengths of 2,766,829,411 and 2,921,040,163 bp (Fig. 2). For further details on the assembly process, see the methods section, Supplementary data 1 and 2, and Supplementary Documents 1 and 2.

BUSCO (v3.0.2)[25] reported 92.6% and 93.1% complete mammalian universal single-copy orthologs in the N'Dama and Ankole assemblies, comparable to the 92.6–93.7% observed across the three previous cattle genomes[24]. Likewise, the duplication levels of 1.4 and 2.1% are comparable to the range of 1.0–1.3% observed across the Hereford, Angus and Brahman genomes. Similarly, the QUAST[26] software (v5.0.2) calculated that the two assemblies cover 93.9% (N'Dama) and 94.0% (Ankole) of the ARS-UCD1.2 Hereford genome, again consistent with the 94.2% and 96.2% of the Angus and Brahman assemblies. Quality values (QV) were calculated using merqury (v1.1)[27] in combination with

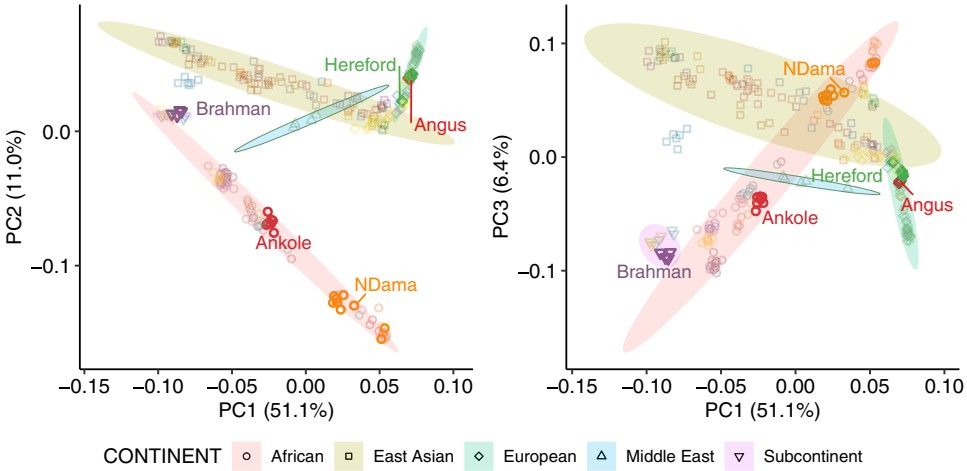

**Fig. 1 Principal component analysis of the 294 cattle.** The positions of the populations of origin of the five assemblies considered in this study are shown. The source data are provided with the paper.

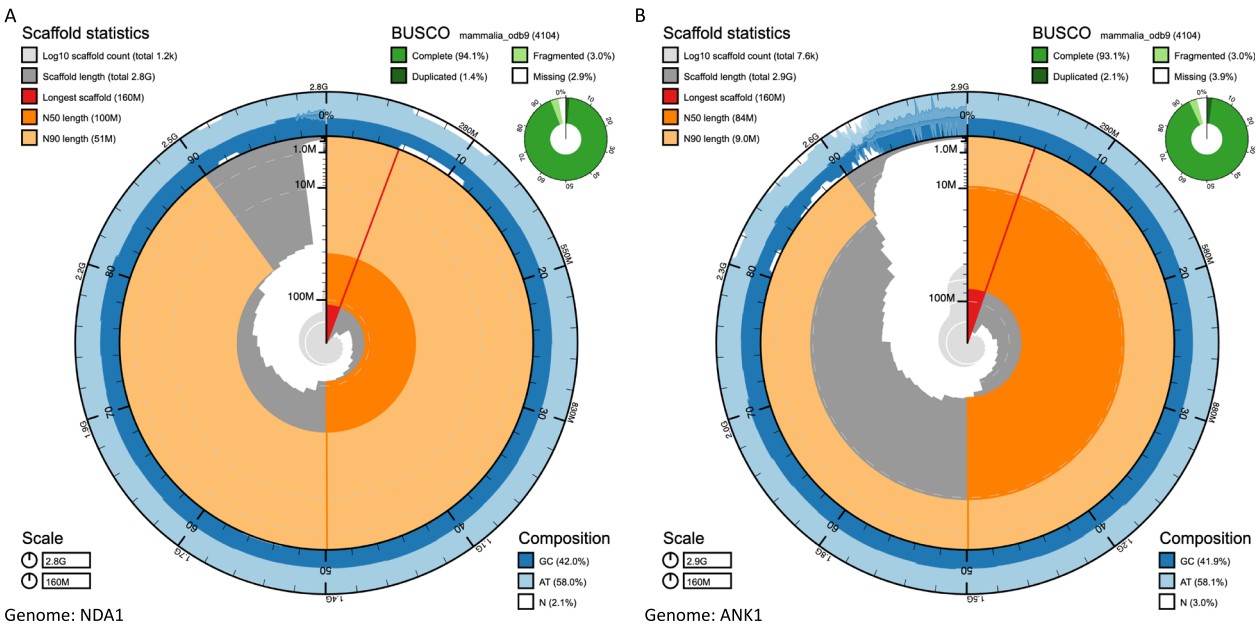

**Fig. 2 Snail plots of the N'Dama (NDA1) and Ankole (ANK1) genome assemblies.** Key metrics are shown for the (**A**) N'Dama and (**B**) Ankole genomes such as the longest scaffold (red vertical line), N50 (orange track), N90 (light orange track), GC content (external blue track) and BUSCO scores (outer circular pie chart in green). The region of elevated N content in the N'Dama assembly corresponds to a 5 Mb gap in one of the contigs matching a region of generalised low identity in all of the five assemblies (Supplementary Fig. 4). Even though this region contained an unfilled gap we observe that the regions flanking the gap align to directly contiguous portions of the genome in other assemblies, and therefore that the gap in this region is potentially smaller than represented here.

meryl (v1.2; https://github.com/marbl/meryl), and were respectively 34.3 (37.9 autosomal) and 30.6 (34.2 autosomal) for the N'Dama and Ankole, with a base accuracy over 99.9%. Finally, RepeatMasker shows that these two genomes share similar contents of the different classes of repetitive elements (Supplementary Fig. 1). These two African cattle assemblies are consequently of good quality (Fig. 2) and represent novel spaces in global cattle diversity. Full details on the assembly processes and their statistics are reported in Supplementary Note 1 and 2.

**Detection of non-Hereford sequence.** We first defined the non-reference sequence present in the non-ARS-UCD1.2 (Hereford) genomes. We aligned the five genomes using the reference-free aligner CACTUS[28], which generates multiple whole-genome alignments (mWGA) in the form of a CACTUS graph. We then converted the graph to PackedGraph format using hal2vg[29] (v2.1), and used a series of custom scripts to extract all the nodes that were not present in the Hereford genome. After excluding nodes encompassing an N-mer, an extra 257.2 Mb of non-Hereford reference sequence across over 29 million nodes was identified (76.7 Mb was from over 23 million nodes in primary autosomal scaffolds; the remaining sequence was on sex chromosome scaffolds or unplaced contigs; Table 1). This value is inclusive of a large number of small nodes, including SNPs, small indels and repetitive elements. Therefore, we excluded all nodes in potentially misassembled regions as identified by FRC_Align[30], combined neighbouring regions (≤5 bp) and filtered out sequences of short length (<60 bp) and those close to a telomere

**Table 1 Sequence contribution from the two African genomes.**

|  |  | Angus | Ankole | Brahman | N'Dama | Total |
|---|---|---|---|---|---|---|
| Non-reference nodes (total) | #nodes | 6,188,973 | 14,994,500 | 14,627,206 | 10,338,166 | 29,315,173 |
|  | bp | 46,066,551 | 118,203,105 | 60,100,791 | 87,792,217 | 257,235,506 |
| Non-reference nodes (autosomes) | #nodes | 5,823,611 | 11,262,561 | 13,362,852 | 8,832,454 | 23,599,013 |
|  | bp | 17,903,582 | 41,317,786 | 39,647,314 | 25,806,882 | 76,660,696 |
| Filtered non-reference nodes (total) | #nodes | 285,307 | 780,815 | 705,024 | 494,781 | 1,008,401 |
|  | bp | 4,612,021 | 12,486,639 | 12,023,827 | 6,760,434 | 15,491,621 |
| Filtered non-reference nodes (autosomes) | #nodes | 198,393 | 429,652 | 443,737 | 313,670 | 571,123 |
|  | bp | 3,290,022 | 7,093,645 | 7,435,063 | 4,595,327 | 9,046,464 |
| Final set of contigs | Number of contigs | 2,250 | 5058 | 6387 | 2970 | 16,665 |
|  | Length (total) | 3,274,775 | 4,508,339 | 10,507,420 | 2,246,905 | 20,537,439 |
|  | Length (min) | 61 | 61 | 61 | 61 | 61 |
|  | Length (max) | 92,590 | 34,789 | 103,683 | 29,488 | 103,683 |
|  | Length (mean) | 1455.00 | 891.00 | 1645.00 | 757.00 | 1,232.37 |
|  | Length (std) | 5177.00 | 1990.00 | 4957.00 | 1885.00 | 3,875.06 |

The table shows the amount of sequences from non-ARS-UCD1.2 genomes, and how much the two novel assemblies from African breeds contribute to the numbers.

or gap, leaving a total of 116,098,017 bp in 62,337 sequences. We further filtered down to sequences that were not significantly more repetitive compared to the average level observed across the autosomes of the different genomes (Bonferroni-corrected $P$-value > 0.05 using a genome-wide mean repetitiveness of 53.99%, see methods for calculation). We finally removed any redundant sequences. This left a total of 16,665 sequences, for a total of 20.5 Mb of high-quality, non-repetitive sequence not present in the Hereford assembly (NOVEL set). The sequences presented a motif content analogous to the genomes of origin, as highlighted by HOMER when using the 5 reference pooled genomes as a background (Supplementary data 3).

The amount of unique and shared sequences within and across breeds is shown in Fig. 3A. The majority of the additional sequence was representative of the indicine ancestry, shared between the Brahman and Ankole, closely followed by the non-Hereford sequence shared across all other genomes, and then from the non-European shared sequence (common across N'Dama, Ankole and Brahman). Of the five breeds, the Ankole genome contained the most non-Hereford sequence (12.4, 7.1 Mb of which resided on primary autosomal scaffolds; Table 1), followed closely by the Brahman genome (12.0, 7.4 Mb on primary autosomal scaffolds; Table 1). A key advantage of multiple genomes is improved representation of divergent loci and Fig. 3B illustrates the divergence between the sequences at the important major histocompatibility complex (MHC). Alignments generated through minimap2 over the whole of chromosome 23 show an identity ranging between 98.77 and 99.31% (for Brahman and Angus, respectively), whereas the 4 Mb interval ranging from 25 to 29 Mb shows an average identity ranging from 96.17 to 98.21%, with local values as low as 43% for some multi-KB fragments (Supplementary Fig. 2).

**Gene content in the novel sequences**. We assessed the NOVEL set of sequences for the presence of genes and gene structures using three complementary approaches (see methods). Blastx alignment identified a total of 191 genes in 272 regions passing the filtering (see "Methods"). Augustus predicted 923 and 1008 genes using the non-reference sequences and the non-reference sequences expanded with 100 bp flanking regions where possible. After filtering out regions that matched, we predicted 182 and 169 using Augustus with and without the 100 bp flanks. Complete genes were then extracted, aligned using BLASTP and genes passing mapping filters were identified for both sets. This identified a total of 132 genes in 158 sequences and 140 genes in 164 sequences in the non-reference contigs and the non-reference

contigs with flanking regions, respectively (Supplementary data 4).

We then combined the resulting 132, 140 and 191 genes from the three methods, and identified a total of 76 genes that were found to be consistent across them. Consistent with their recent origin, most of these genes represented multi-gene families including several predicted immune genes (e.g. Ig lambda chain V-II region MGC, interferons alpha and T-cell receptor beta chain V region LB2), melanoma-associated antigens (*MAGEB1*, *MAGEB3* and *MAGEB4*) as well as a number of olfactory receptors (Supplementary data 4).

**Constructing the graph**. We next assessed the potential of using these new assemblies as part of a graph genome. To enable the comparison of graph-based variant calling performance, four versions of vg-compatible genomes were generated (a schematic representation of these can be seen in Fig. 4A). The first contained the Hereford genome only (which we refer to as VG1). The second was VG1 augmented with 11,215,339 million short variants called across 294, largely unrelated, animals (Fig. 1) from a globally distributed selection of cattle breeds[23] (VG1p). The third contained all variants (SNP, small InDels and SVs) derived from the five cattle assemblies (VG5), and the fourth contained all variants from the five assemblies previously described and augmented with the over 11 million variants (VG5p). We also constructed a version of VG1p prioritizing 754,144 variants for the three breeds considered in the downstream analyses (Angus, N'Dama and Sahiwal) using FORGe[31] (see "Methods" for the details of the prioritization). However, due to the very modest change in the reads mapped when compared to the VG1p graph (see Supplementary data 5), we did not take this version forward for downstream analyses.

The graph genome based on the CACTUS alignment only (VG5) had an order of >147 million nodes (i.e. the number of fragments of sequences) and a size of >173 million edges (i.e. the number of connections between nodes), doubling the order of the linear graph produced using just the autosomal sequence of the Hereford genome (VG1), that had >77 M nodes and edges (Supplementary Table 1). Including the genetic variants from the 294 cattle led to >105 M nodes for VG1p and 163 M nodes and 194 M edges for VG5p (10% more nodes and 12% more edges than VG5).

**Read mapping to linear and graph genomes**. To assess the performance of these genome versions we aligned short-read sequencing data from nine animals spanning three diverse breeds

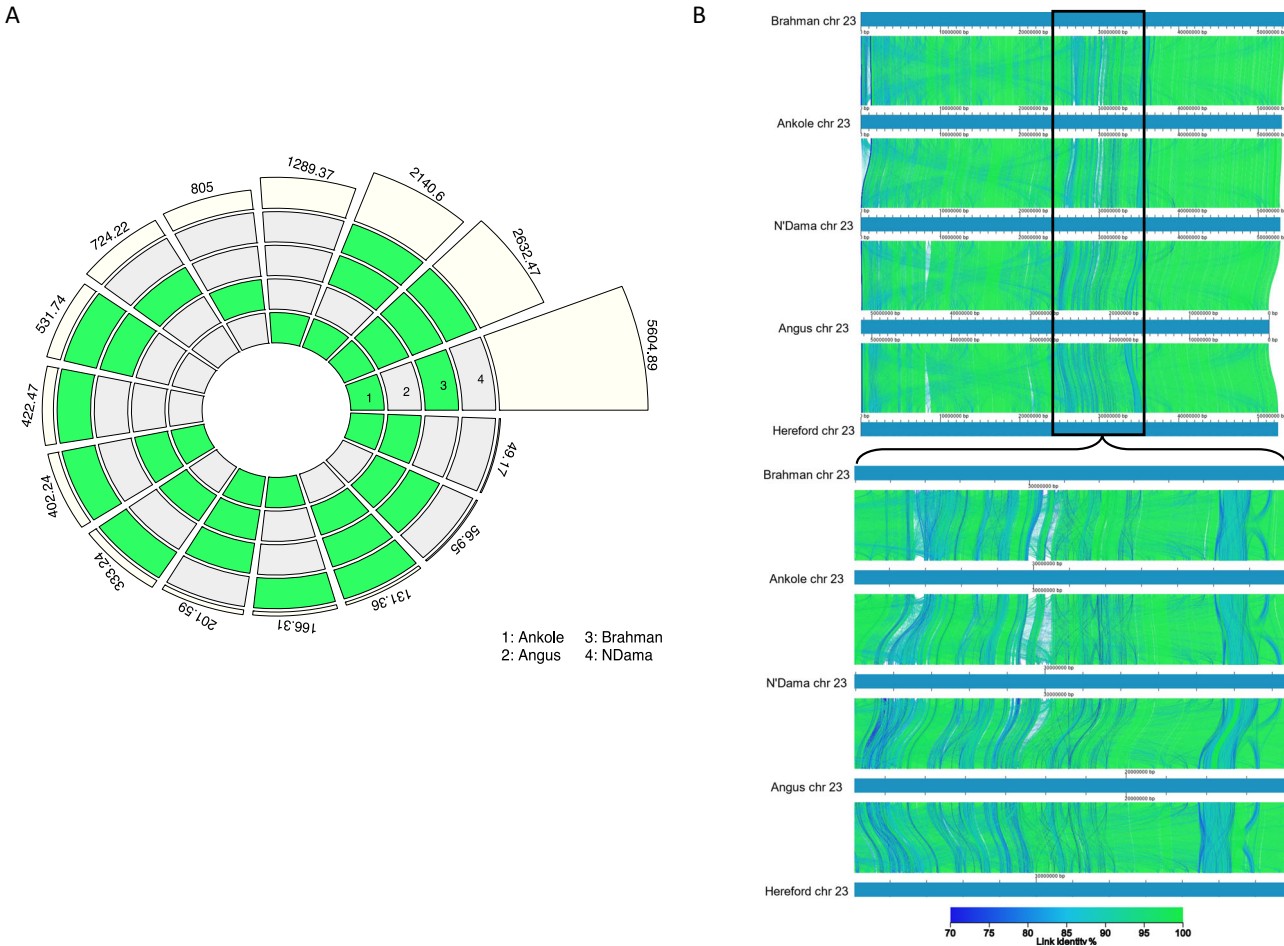

**Fig. 3 Comparison of genomic content across the genomes. A** High-quality (NOVEL) sequence specific to, or shared among, each non-reference genome. Numbers represent the kilobases of non-Hereford sequence associated with the set of genomes defined by the group(s) highlighted in green. Each genome is indicated by a number (1 = Ankole, 2 = Angus, 3 = Brahman and 4 = N'Dama); **B** Multiple genome alignments of the MHC region on chromosome 23 generated with AliTV (v1.0.6)[75]. The plot represents the shared sequences among the different genomes; blue to green segments are representative of higher to lower similarity (100 to 70% respectively); the enlarged region is the MHC region, which shows a large amount of variation between the assemblies.

(three European taurine Angus animals, three African taurine N'Dama and three indicine Sahiwal) to each version. Importantly, genotypes from these animals had not been included when constructing the graphs. An advantage with graph genomes is in theory they should increase the number of reads directly matching a route through the graph and, consistent with this, we observed between 10 and 27% more reads perfectly mapped with vg to the CACTUS graph representation of the cattle genome (VG5) than to the Hereford only version (VG1) (Fig. 4B). The greatest increase in perfect read mapping was for the indicine Sahiwal breed, followed by the N'Dama and finally the Angus animals, mirroring the relative divergence of each from the Hereford breed. A modest further improvement was observed when aligning to the full graph incorporating the short variant data (VG5p) (an extra 0.52% of perfectly mapped reads among the Angus to 3.25% among the Sahiwal). Although direct comparisons across different software tools are difficult and need to be treated with caution, we found that vg aligned 7–10% more reads to the graph than BWA to the primary chromosomal scaffolds of the ARS-UCD1.2 (Supplementary data 5).

**Variant calling from linear and graph genomes**. We calculated several key metrics to describe the variants called using VG, GATK and FreeBayes, and collected them in Supplementary

Note 3, both considering the fixed set of 11 M variants as "known" variants (case A) and considering the variants used to construct each graph as "known" (case B). These plots show how the variants called using the three algorithms (VG, FreeBayes and HaplotypeCaller) presented similar quality, depth, number of variants, mapping quality and, generally, comparable metrics when looking at depth of sequencing (DP), quality of the variants and number of variants called (Supplementary Note 3).

A key metric when assessing the quality of read alignments to a genome is allelic balance (AB). Ideally, reads carrying each allele at a polymorphic site should be equally well mapped to the reference genome (i.e. have an AB = 0.5). In practice though, there is usually a bias towards reads matching the sequence present in the reference genome at the location. Skewed allelic balance can adversely affect variant calling and therefore reducing it can improve downstream genetic analyses. The allelic balance observed across genomes, variant sizes and types is shown in Fig. 4C, with alternative representations which consider all the types of graph considered shown in Supplementary Note 3. Consistent with previous studies in humans, this figure illustrates that the allelic balance at short variants is generally comparable for single nucleotide polymorphisms, and the allelic balance at small InDels (<15 bp) doesn't show a particular improvement compared to variants called using standard variant callers.

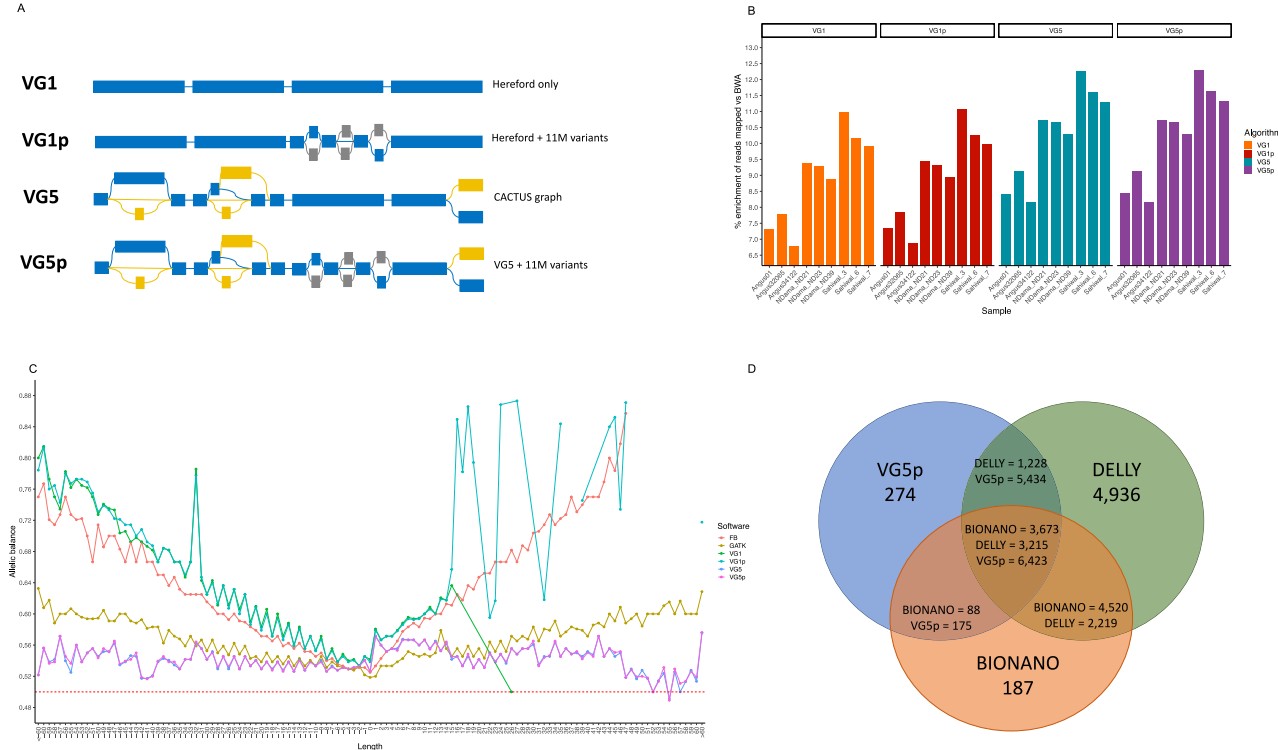

**Fig. 4 Graph genome descriptions and their performances. A** A cartoon representation of the four types of graph genomes considered (the linear VG1, VG1 expanded with 11 M short variants (VG1p), the CACTUS VG5 graph and the CACTUS graph expanded with the 11 M short variants (VG5p)). Regions indicated in blue are regions coming from the backbone sequence, those in grey are the short variants from Dutta et al. (2020), and in yellow the variants derived from the CACTUS graph; **B** the percent enrichment of reads mapped by vg (primary axis) using the different graphs over the bwa mem linear mapper; and **C** the allelic balance for the linear callers FreeBayes and GATK HaplotypeCaller compared with vg call, showing how the latter reduces the allelic bias for large variants. For other versions of this plot looking at different sets of known and novel variants see Supplementary Note 3; and **D** the intersection of structural variants longer than 500 bp called using the VG5p graph (blue), Delly V2 (green) and the Bionano optical mapping (orange), showing how most variants called with vg are also confirmed using one of the other methods. Note an SV called by one method may overlap more than one SV called by a different method. The source data for panels (**B**), (**C**) and (**D**) are provided with the paper.

However, calls from the graph shows an overall better allele balance for larger variants (>15 bp long) than both GATK and FreeBayes, staying closer to the desirable value of 0.5 (Supplementary Note 3). Defining the variants as known if used when constructing a particular graph allows for a less uniform comparison, but still confirms the ability of the graph to call larger variants with an overall better allelic balance than the standard variant callers (Supplementary Note 3). Interestingly, while marginally more reads were successfully mapped to the VG1p graph than to VG1, it displayed a less consistent allelic balance at insertions between 10 bp and 40 bp long. The best results were achieved using the VG5p graph, though with the largest gains observed in VG5 vs VG1 and VG1p, highlighting the benefits of the additional assemblies in the graph (Supplementary Note 3).

We also evaluated other metrics for the different approaches, including DP, average quality of the call (QUAL), number of variants called, transition/transversion rate (Ti/Tv), that are presented in Supplementary Note 3. Overall, the metrics for the VG graphs look similar to the classical callers, with just the Angus sample from public databases presenting a lower Ti/Tv ratio.

**Assessment of graph genome structural variant calls.** One of the most important benefits of graph genomes is the ability to directly detect large variants using short-read sequencing data. Using the VG5p graph genome we were able to genotype thousands of structural variants of 500 bp or longer, i.e., longer than the length

of the reads being mapped (Supplementary Note 3). These SV regions are inaccessible and uncalled using linear callers such as GATK or FreeBayes, making vg a suitable tool for explicit genotyping of large variants. To assess the quality of these SV calls, and to test its utility when applied to the study of African breeds, we compared the variants called on the VG5p graph to independent Bionano optical mapping (OM) data for two additional N'Dama samples. As OM is a distinct technique for identifying the location of SVs, based on staining and imaging large DNA fragments, it provides an independent indication of SV location. It should be noted that the N'Dama used for whole-genome resequencing and the OM were from completely different countries (Nigeria and Kenya, respectively) though the OM data and N'Dama assembly was from animals from the same research institute.

In total, vg detected 12,306 structural variants of >500 bp across the nine samples, each of which might have one or more alleles per region. Of these, 6598 overlapped with regions detected by the Bionano OM data. Despite the comparison with OM data of one breed only, this number is ~3.4 times higher than expected from randomly selecting sections of the genome of the same size (mean ± standard deviation of $1571.2 \pm 36.9$ across 10,000 permutations; Z-score = 136.1, $P < 2.2 \times 10^{-16}$; Supplementary Table 2). Further supporting the validity of the indel calls, in-frame indels called from the graph were observed to be more common than other coding indels, consistent with selection disproportionately removing frameshift changes (Supplementary Fig. 3).

Consistent with the OM data being deriving from the same breed, the number of graph SVs >500 bp overlapping the OM SV calls was greatest in the taurine N'Dama (2932/7280, 40.3%; average size 2055.4 bp), followed closely by the taurine Angus (2797/7318, 38.2%; average size 2050.7 bp) with the lowest overlap with the indicine Sahiwal (3368/10,046, 33.5%; average size 1880.9 bp; Supplementary Table 3). Again, the number of variants detected in each different breed is reflective of the distance from the reference genome considered.

We detected 19, 49 and 299 high-quality, large structural variants found across all Angus, N'Dama and Sahiwal samples, respectively, but not in the other breeds (i.e. that were specific for a breed and with QUAL > 30, 20 < DP < 90, alternate allele count ≥ 5, >500 bp). These SV are therefore common to a given group but not found across breeds, and the numbers likely reflect the relative genetic divergence of each breed from the Hereford genome used as the backbone for the graph.

To confirm the quality of these variants, we overlapped them with the N'Dama OM data. Results for each breed are shown in Supplementary Table 2. Despite the OM data being derived from different individuals, there was a substantial overlap between the N'Dama SV calls, with 42 out of 49 overlapping across both approaches (85.7%), much more than the number of overlaps expected by chance (mean ± standard deviation of 6.2 ± 2.3 on 10,000 repetition; $Z$-score = 15.3, $P$-value = $1.40 \times 10^{-52}$; Supplementary Table 2). Although the overlap between the N'Dama OM and Angus and Sahiwal graph SV calls was lower, both showed a significant overlap (10/19; 52.6% and 111/299; 37.1%, respectively; Supplementary Table 2) The partial overlap with these breeds may reflect that not all of these SV are actually breed specific but rather are just more common in the breeds, or potentially the comparatively low resolution of the OM data results in false positive overlaps. Either way a much higher overlap is observed with the N'Dama SV calls, consistent with these group-restricted calls being much more enriched in this population, and consequently the genome graphs appear effective at identifying these larger SV.

**Comparison with Delly SV calls**. Next, we compared the results from VG5p with structural variants called through a classical SV caller, Delly (V2), using the linear Hereford genome as the reference. After excluding SVs with low depth, imprecise positioning and translocations, we found on average 7218 variants for the Angus (6878 to 7533), 15,978 for the N'Dama (15,061 to 17,399) and 30,856 for the Sahiwal samples (30,466 to 31,162) as shown in Supplementary Table 4. These SVs were combined using SURVI-VOR (v1.0.7) merging SV regions if <100 bp apart when accounting for the SV type. SVs were further filtered to those with at least 1 sample supporting it and with a size >500 bp to make them broadly comparable to the OM data given the latter's resolution (Supplementary Table 4). This filtering excluded all the insertions, since Delly is incapable of calling insertions with precise break points, limiting the types of SV analysed to deletions, duplications and inversions. The filtering left 3175 unique SVs for the Angus (ranging from 1940 to 2167 genotyped in each samples), 5206 unique SVs for the N'Dama (ranging from 2945 to 3418 genotyped in each samples) and 8421 unique SVs for the Sahiwal samples (ranging from 5356 to 5396 genotyped in each samples).

In total, 11,562 precise non-translocation Delly SVs with suitable depth and size were retained across all individuals. Of these less (5371, 46.4%) overlapped with an SV called from the OM data than for vg (6598, 53.6%) (Supplementary Table 4). Therefore, from the same sequencing data, more SVs were called using vg that were also more likely to overlap an SV called from the independent OM data.

Figure 4D shows how the structural variants called by vg are confirmed by at least one of the other methods, with only 274 out of 12,306 remaining unsupported (2.2%). In contrast Delly called 4936 SV unsupported by either other method. It should be noted though that Delly called 2219 SVs overlapping an SV in the OM data not identified by vg. These are potentially sample-specific SVs, that being absent from the graph will be largely uncalled by vg. Further improvements to the graph, for example by including further assemblies, would be expected to reduce this number.

Finally, when looking specifically at deletions, the only class in common among the three methods, we find that Delly calls a higher raw number of SVs compared to vg, detecting 3186 deletions with a match in the OM data, whereas vg calls 1887 SVs with overlaps. This higher number of SVs called by Delly is probably reflective of its ability to take advantage of split reads and unaligned reads to identify novel SVs in the sample. However, in proportion to the number of deletions called by each, Delly has a lower proportion of confirmed SVs (3186/9030 = 35.3%) than VG (1887/3972 = 47.5%), highlighting the higher specificity of the graph approach.

An example of a high-quality 1530 bp sequence absent in the Hereford genome, but present in the graph, is in an intronic region of *HS6ST3* (Heparan-sulfate 6-O-sulfotransferase; hereford.12: 73,579,158, Fig. 5). This SV was identified by both OM samples (Fig. 5A), the three re-sequenced N'Dama genomes (Fig. 5B) and was present as an alternate sequence in the graph but not identified by Delly, even without filtering any SVs from the different samples (Fig. 5C).

In conclusion, assembly-based graphs are a viable solution for reliably calling SVs with explicit alleles, including insertions that are generally of lower quality in classical SV callers. Future additions of new breed-specific reference assemblies would be expected to further improve the number of variants represented in these graphs, ultimately improving the structural variant calling and analysis.

**ATAC-seq peak calling**. After analysing variant calling on the graph genome, we tried to investigate whether other omics analyses may also benefit from these novel resources. To do so, we obtained ATAC-seq data for three animals belonging to the three main clusters of cattle diversity: European taurine (1 Holstein-Friesian), African taurine (1 N'Dama) and indicine (1 Nelore), plus a nucleosome-free DNA as an input sample to remove likely false positive peaks.

Peak calling directly from graph genomes is currently an under-developed field, with ongoing issues in supporting graphs inclusive of large variants; therefore, in the short-term, studies of chromatin and the epigenome are likely to continue to use linear genomes. We consequently took advantage of the NOVEL set of high-quality non-reference sequences described above to create an expanded version of the current linear genome we term here ARS-UCD1.2+. This expanded genome contained in total an additional 16,665 contigs across the over 20 Mb of sequence, with a mean length of 1.23 kb (S.D. 3.87 kb and a range of 61 to 103,683 bp long Table 1). This increased the reference size by 0.7% to 2780 Mb.

To explore the potential benefits of these new data to such analyses we aligned the reads and called the peaks for each sample separately to the five different linear genomes, as well as the expanded ARS-UCD1.2+. We aimed to minimise the impact of multi-mapping reads (see "Methods") and after calling peaks, we excluded all peaks shared with the input sample for more than 50% of their length.

Figure 6 shows using the ARS-UCD1.2+ genome leads to a modest increase in the number of peaks called relative to the standard Hereford ARS-UCD1.2 sequence (Supplementary

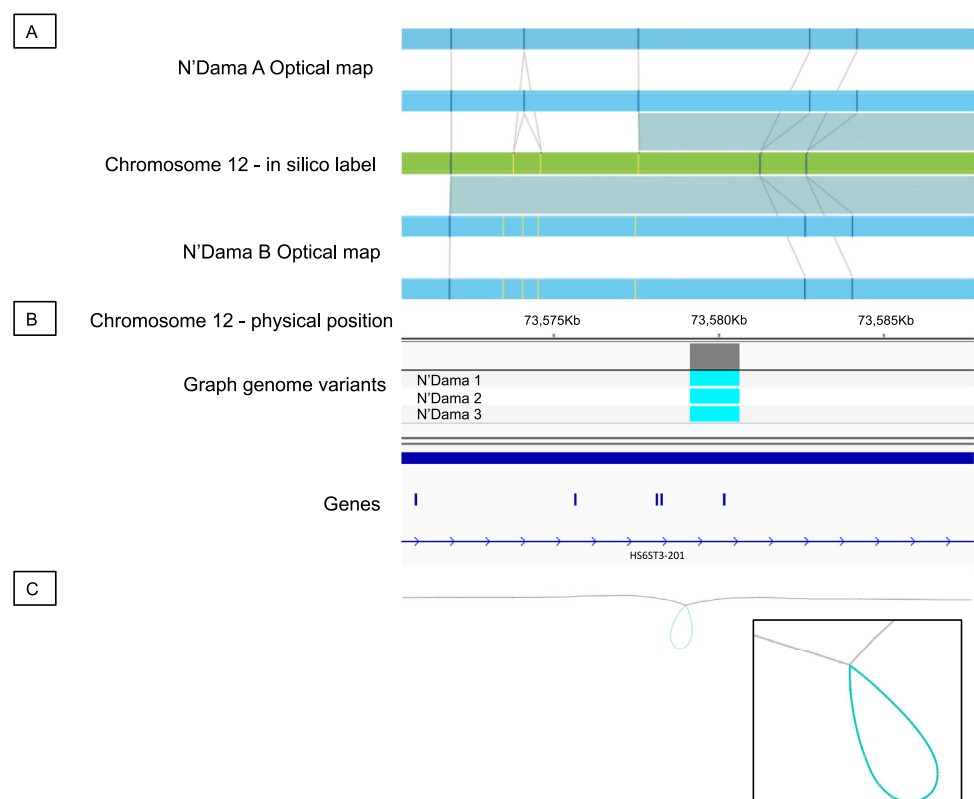

**Fig. 5 Example of an insertion in the N'Dama relative to the Hereford reference.** The insertion was detected **A** in both Kenyan N'Dama OM samples as represented by an increase in the distance between labels (vertical lines) on each bionano haplotype (blue rectangles) over that expected given the labels' in silico locations in the Hereford reference (green rectangle). **B** This SV was identified as homozygous in all three Nigerian N'Dama resequenced genomes when called against the graph genome. **C** A Bandage[76] representation of the graph genome in this region showing the large structural variant (blue loop) in the Hereford genome (grey line).

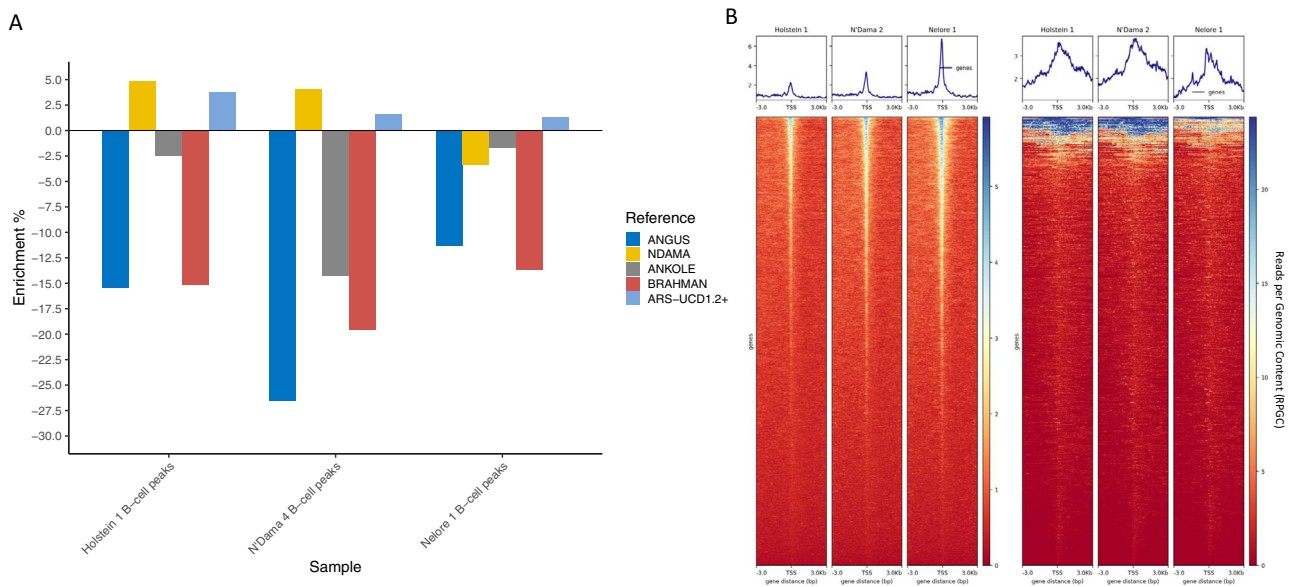

**Fig. 6 ATAC-seq analyses results. A** Enrichment or depletion of the number of ATAC-seq peaks called in the different assemblies with respect to the number called in ARS-UCD1.2, showing more peaks were called using the expanded ARS-UCD1.2+ genome in all samples; and **B** showing the enrichment around the TSS of both the ARS-UCD1.2 annotated genes (left three heatmaps) and of the 923 features predicted by Augustus in the novel contigs (right). The source data for panel (**A**) are provided with the paper.

Table 5). This increase is confirmed also when using only uniquely mapped reads, with the ARS-UCD1.2+ calling consistently more peaks than the standard ARS-UCD1.2 (Supplementary Table 6).

Peak calling on the ARS-UCD1.2+ genome returned up to 3.7% more peaks when compared to the ARS-UCD1.2 genome at the same significance thresholds despite ARS-UCD1.2 being only 0.7% longer. This expanded genome worked particularly well for

the Holstein, which generally showed a higher number of peaks called compared to the ARS-UCD1.2 assembly (+3.7% peaks called), followed by the N'Dama sample, with an extra 1.6% of additional peaks called and finally the Nelore (+1.3% peaks called; Fig. 6A and Supplementary Table 6). Intersecting these novel ATAC-seq peaks with the predicted genes in the 20.5 Mb of non-Hereford (Supplementary data 6), non-highly repetitive sequences identified a general enrichment around their predicted TSSs, consistent with these novel peaks marking regulatory elements uncaptured by the Hereford genome (Fig. 6B). Over 93–96% of these peaks matched a peak in the genome of origin (i.e. a peak called on a non-reference sequence from the Angus genome has a matching peak on the Angus genome in the same region), further supporting the potential content of functional elements (Supplementary Table 6).

Consequently, the use of more representative pan-genome resources likely has utility to downstream analyses beyond just variant calling, including identifying the location of novel regulatory elements missed when using current reference resources.

## Discussion

In this study we generated the first two cattle reference genomes of African taurine and Sanga (an ancient stabilized cross between indicine and taurine breeds[32]) lineages. These two new sequences have quality metrics comparable to those of other currently available reference genomes, and will likely be important resources for future bovine genomic studies, in particular those studying non-European breeds.

By aligning the five cattle assemblies, we illustrate that a substantial portion of the cattle pan-genome is likely missing from the Hereford reference. The amount of non-reference sequence identified by our approach broadly matches that from another study using a different but overlapping set of genomes and graph assembly approach[33]. This has important implications for cattle research as it suggests significant amounts of the bovine genome is inaccessible in most current analyses. Although a proportion of this extra sequence is repetitive, unsurprisingly given its recent origins and the simple fact that large parts of mammalian genomes are made up of repeats, this does not preclude it from being functional. For example, the importance of repetitive elements in gene regulation is becoming increasingly clear[34]. Consequently, the study of these DNA segments that are not common to all animals may provide further insights into the drivers of phenotypic diversity between breeds.

One noteworthy observation was that the amount of extra sequence in each genome matched the prior assumptions of the relationships between the breeds: the two indicine genomes (the Ankole and Brahman) had the highest amounts of unique, non-repetitive sequence. Considering that the sequences identified might contain functional elements as predicted by our analyses, there is the case for sequencing more genomes from the most distantly related lineages from the reference Hereford assembly, such as the *Bos indicus* lineage, since they might contribute further additional functional regions.

In this study we illustrate that the use of the graph cattle genome does not lead to substantial improvements in the calling of SNPs and small indels, even when large numbers of them are integrated into the graph. This is likely reflective of the relative maturity of short variant callers such as GATK which are already highly accurate and which already effectively adopts a simplified, localised graph approach by constructing and aligning to plausible haplotypes at polymorphic regions. Previous studies have shown that the prioritization of variants included in the graph can potentially lead to further improvements in alignment

accuracy[17,31]. Although this approach has proven powerful for population-specific graphs, its use for pooled populations or less standardized breeds is less clear, and comes with the disadvantage that different genomes are used in different analyses. A key aim of this project was to generate a more representative, non-breed specific, genome more relevant to researchers studying the diverse set of often admixed breeds across low and middle-income countries.

While variants calling of SNPs and small InDels is appropriately addressed by standard algorithms, graph genomics improves both read mapping and calling of larger structural variants thanks to the presence of non-reference sequences across the different assemblies. Arguably, neither GATK Haplotype-Caller nor FreeBayes is a structural variants caller, and this function typically requires specialised tools such as Delly[35]. Our analyses show how the structural variants called using a multi-genome graph are more consistent with SVs called using independent OM data than those from Delly, with over 53% of SV called from a graph genome overlapping an SV region called from OM data whereas the SV called through Delly overlap 46% of the time. When looking specifically at overlapping deletion calls these numbers were 48% and 35% respectively. Importantly, whereas tools such as Delly struggle to accurately call SVs such as insertions from linear references, graph genomes enable these to be accurately genotyped where present in the graph. The greater the diversity present in the graph, the better SV calling will become. Unlike linear genomes whose content is largely fixed. Reassuringly, SVs called among N'Dama samples using the genome graph were more consistent with N'Dama OM data than the SV called in other breeds. Although a perfect overlap would not be expected given different animals were being studied, the overlap among the N'Dama was 86% compared to 37% among the more distantly related Sahiwal. It should be noted that Delly was able to call a large number of potential SVs not present in the graph. Further augmenting the graph with more genomes should though reduce this number.

In comparison to linear reference genomes there are currently few viable software tools for epigenetic and chromatin analyses using graph genomes. However, using ATAC-seq data across breeds we demonstrated it is possible to call substantially more peaks using an expanded version of the linear reference genome incorporating the extra sequence found in the other genomes. When applying the same thresholds and accounting for multi-mapping reads, 3.7% more peaks were called across Holstein-Friesian ATAC-seq datasets compared to using the standard linear reference. This is despite the expanded reference only being 0.7% longer, and not less than 1.3% of extra peaks being called on each individual considered. Although the use of pan-genomes to study chromatin is a particularly immature field, pan-genomes have the potential to reduce noise due to the more accurate representation of structural variants and large rearrangements.

When looking across the results of both structural variants calling and ATAC-seq peak analyses, we can see that our genomes work well, and in particular for breeds present or closely related to ones used to generate the graph and expanded genome, highlighting the need to increase the genetic diversity that underpins the graph, particularly for lineages that are poorly represented.

Despite these improvements, graph genomes still have drawbacks. These methods are still under active development, and still have a greater requirement of computer memory, disk space and analytical time. Generating a whole-genome assembly is time consuming, generating the vg graph itself still requires large amount of memory (up to several terabytes), and still can only be done on primary chromosomal scaffolds due to high storage demands. Alignments are also more computationally intensive

than with their linear counterparts, with the requirements affected by the number of variants represented. Moreover, variant calling currently relies on a pile-up approach, which is arguably less sophisticated than methods implemented by GATK or FreeBayes, that likely helps explain the good performance of traditional tools at calling SNPs and small indels[36]. Methods for peak calling on graph genomes are not always compatible with graphs generated through CACTUS or similar software, which limits their application and was one of the stimuli for generating the ARS-UCD1.2+ genome. Last but not least, although efforts are being made to resolve the coordinate system for graph genomes, downstream analyses are more complicated due to most current resources being referenced to the positions on one linear genome.

Nevertheless, it is clear graph genomes already have advantages in certain areas such as SV calling. As the field of graph genomes is less mature, arguably there is greater scope for further improvement. New genomes are being released at a much higher frequency than in previous years, and initiatives such as the recently announced bovine pangenome project[37] will open new possibilities and allow a better understanding of cattle genetics and phenotypic diversity.

We consequently present the first African cattle genome assemblies integrated into a cattle graph genome representing global breed diversity. This graph, incorporating both large SVs and millions of SNPs from across global breeds, is demonstrated to improve downstream analyses such as SV calling and the detection of novel functional regions and therefore has the promise to improve our insights into the genomics of this important livestock species.

## Methods

**African breed assemblies**. Whole blood of the N'Dama bull N195 was collected in PAXgene DNA tubes. The bull was located at ILRI's Kapiti research station in Machakos county, Kenya. The PAXgene DNA tube was stored at room temperature overnight and then the fridge at 4 °C for 1 day prior to DNA extraction. The standard procedure was used as outlined in the PAXgene blood DNA kit handbook. Resulting DNA was sequenced using the Pacific Biosciences (PacBio) Sequel platform at Edinburgh Genomics, yielding a total of 13 M reads and 109 Gbp, corresponding to a genomic coverage of ~40X. In addition to long reads, the same animal was re-sequenced using Illumina HiSeq X Ten paired-end short-read (PE-SR) sequencing, yielding 260Gbp with an average insert size of 250 bp, corresponding to a genomic coverage of ~80X.

A whole blood sample of the Ankole bull UG833 was collected in PAXgene DNA tubes from a farm in Uganda, and DNA was extracted using the same protocol described for the N'Dama sample. It was then sequenced by Dovetail genomics using the Pacific Biosciences Sequel sequencing platform which yielded a total of 10 M reads and 107Gbp, corresponding to a genomic coverage of ~38X. The same animal was re-sequenced using Illumina HiSeq X Ten paired-end short-reads, yielding 260Gbp with an average insert size of 250 bp, corresponding to a genomic coverage of 60X. Finally, OM samples were prepared starting from monocytes using blood collected by jugular venupuncture into EDTA vacutainers. Following erythrocyte lysis monocytes were purified from the leukocytes using a positive selection MACS protocol with an anti-bovine SIRPα mono-clonal antibody (ILA-24, Roslin Institute purified stock (lot 14), concentration 1 μg/ml[38]. Agarose plugs containing $5 \times 10^5$–$1 \times 10^6$ isolated monocytes were prepared using the Bionano Blood and cell culture DNA isolation kit (Bionano Genomics, San Diego, US) according to the manufacturer's instructions and the extracted DNA used for analysis on the Bionano Saphyr platform. The procedure yielded 3.5 M molecules with an N50 of 245.25 Kbp and spanning a total length of 611 Gb, corresponding to 120X haploid genomic coverage.

DNA from Uganda was received under a license from the Uganda National Council for Science and Technology (permit number A579). All protocols involving animals were approved prior to sampling by the relevant institutional animal care and use committee (ILRI IACUC or Roslin Institute Animal Welfare Ethical Review Body). All blood sampling was carried out by trained veterinarians, according to the approved institutional protocols.

**N'Dama assembly**. Briefly, N'Dama long reads were assembled testing both the CANU (v1.8.0)[39] and FALCON-Unzip pipeline (v1.2.5)[40], keeping the assembly with the highest contiguity. The assembly generated with FALCON was retained due to presenting the highest contiguity and polished twice using minimap2-mapped (v2.16-r922)[41] long reads and the racon (v1.4.3) software[42], and then

further polished once using Pilon v1.23[43] and the 80X of short reads. After that step, contigs were aligned to the three high quality cattle reference genomes (ARS-UCD1.2 [http://bovinegenome.elsiklab.missouri.edu/sites/bovinegenome.org/files/GCF_002263795.1_ARS-UCD1.2_with_y_refseq_chrids.fa.gz], UOA_Brahman_1 [https://www.ncbi.nlm.nih.gov/assembly/GCF_003369695.1/], UOA_Angus_1 [https://www.ncbi.nlm.nih.gov/assembly/GCA_003369685.2/] representative of Hereford[4], Angus[24] and Brahman[24], respectively) using SibeliaZ (v1.1.0)[44] and then scaffolded into chromosomes with Ragout2 (v2.1.1)[45] allowing for the break of chimeras, and processing separately the autosomes, mitogenome, X, Y and the remaining contigs (Supplementary Note 1). Briefly, autosomes have been assembled using the complete set of polished contigs and considering the autosomes from the Angus, Hereford and Brahman genomes as references. Then, we identified the mitochondrial genome by aligning the unscaffolded contigs with the Hereford mitogenome, and fixed misassemblies manually. The remaining unplaced fragments have then been used to scaffold the sex chromosomes. By using the same set of contigs we tried to (a) overcome the limited number of reference sexual chromosomes available (X from Hereford and Brahman, and Y from Hereford and Angus) and (b) address the pseudo-autosomal regions. Then, fragments unplaced in both X and Y were collected and used to identify the N'Dama specific sequences by comparing them to the remaining contigs from the three reference genomes (for details on the reference-assisted scaffolding, see Supplementary Note 1). Although an alternative strategy to scaffolding this genome would have been to use Bionano data from its offspring we did not find using this approach substantially altered the genome or the conclusions of this study. Unlike Ragout, the Bionano scaffolding did not successfully generate chromosome-level scaffolds in all cases, and we estimated that using the OM approach would lead to <30 Kb of N'Dama-specific sequences being altered among the primary scaffolds.

Following the generation of chromosomes, we proceeded with the gap filling through LR_GapCloser (v1.1)[46], using the PacBio long reads and performing three mapping and filling iterations with chunks of 300 bp. Finally, the assembly has been polished five times using Illumina PE-SR and the Pilon v1.23 software. By keeping tracks of the changes introduced by each polishing it was possible to define at which step to freeze the genome version. Resulting assembly statistics are show in Table 1: after the scaffolding, there was a minor reduction of the contig N50 due to some contigs being found to be chimeric and, therefore, fragmented at the breakpoints. However, gap filling and subsequent polishing increased the N50 of the contigs to >10 Mb, confirming the high contiguity of the assembly. Scaffold N50 and L5 are 104,847,410 bp and 11, respectively. Several quality metrics have been collected, such as BUSCO (v3.0.2)[25] completeness scores, QUAST (v5.0.2)[26] evaluations, Merqury (v1.1)[27] QV and FRC_Align (v1.3.0)[30] to identify the candidate misassembled regions. Key metrics (N50, L50, longest contigs, number of contigs, GC content, BUSCO scores) have been represented as SnailPlots using BlobToolKit (v2.3.3)[47]. Details of the assembly, with all the steps performed, is reported in Supplementary Note 1.

**Ankole assembly**. The Ankole long reads were assembled using both the WTDBG2 (v2.3) ultra-fast assembler[48] and CANU[39]. Both sets of contigs were polished twice using minimap2-mapped long reads and the wtpoa-cns software[48]. Then, to overcome the differences that can be produced by the two assemblers, contigs from both software were joined using quickmerge[49] (v0.3; parameters -hco 15.0 -c 5.0 -l 2,500,000 -ml 50,000). This generates a set of contigs with a four-fold improvement in contiguity. The scaffolding step was performed on this set of molecules using the OM data and the Bionano Solve assembly and hybrid scaffolding pipelines, which has the additional advantage of detecting and fixing eventual chimeras introduced by the assemblers and quickmerge pipelines.

Following the generation of chromosomes we proceeded with the gap filling through LR_GapCloser[46], using the PacBio long reads and performing three mapping and filling iterations with chunks of 300 bp. The gap filled assembly was polished five times using Illumina PE-SR and the Pilon software (v1.23). The same metrics collected for the N'Dama assembly have been used to freeze the genome version. Several quality metrics have been collected, such as BUSCO[25] completeness scores, QUAST[26] evaluations, Merqury[27] QV and FRC_Align[30] to identify the candidate misassembled regions. Key metrics (N50, L50, longest contigs, number of contigs, GC content, BUSCO scores) have been represented as SnailPlot using BlobToolKit[47]. Details of the assembly, with all the steps performed, is reported in Supplementary Note 2.

**Genome alignment and comparison**. We compared the five genomes by first generating mWGA using CACTUS[28] (v2019.03.01, installed through bioconda). CACTUS is a mWGA tool allowing reference-free comparison of multiple mammalian-sized genomes. The software requires only the soft-masked genomes (soft-masking largely decreases the computational time) and a phylogenetic tree defining the relationships among the genomes analysed used to guide the alignments.

We masked repetitive elements inside the assemblies using sequentially DustMasker (v1.0.0 from blast 2.9.0)[50], WindowMasker (v1.0.0 from blast 2.9.0)[51] and finally RepeatMasker (v4.0.9, with trf v 4.09)[52]. The reports generated by RepeatMasker on repetitive element composition for the different sequences have been collected using an in-house script and summarized in Supplementary Figure 1. Then, we generated a tree inclusive of the different cattle breeds using mash (v2.2)[53] on a broader set of genomes, inclusive of water buffalo

(UMD_CASPUR_WB_2.0)[54], goat (ARS1)[55], sheep (Rambouillet_1.0), horse (EquCab3.0) and pig (SScrofa_11)[56] in order to achieve a more stable tree and extracting from that the specific branch of interest.

Following the generation of alignments with CACTUS, we used a custom pipeline to detect nodes that were not present in the Hereford genome, ARS-UCD1.2, considered as the reference genome. We first used a custom python script and the libbdsg[57] library to extract the nodes not present in any Hereford paths. These nodes have then been screened for N-mers, and then misassembled regions detected by FRC_Align[30] on the two de novo assemblies here presented were discarded. Each node passing the filtering has been labelled depending on which path it was found. We then combined regions that were <5 bp apart using bedtools (v2.30.0)[58], and classified depending on their length (short if <10 bp, intermediate if between 10 bp and 60 bp and large if ≥60 bp), position (telomeric if within 10 Kb from the end of the chromosome and flanking a gap if with 1 Kb of a N-mer), type of sequence (non-reference if >95% of the bases in the region are not present in any Hereford node, haplotype otherwise). We then added the proportion of masked bases in the regions generated. We then applied multiple filtering to retain only the high quality non-reference contigs, keeping a region if (1) classified as large, (2) consisting of more than 50% non-reference bases, (3) not telomeric, (4) not flanking a gap and (5) not significantly enriched for repetitive elements (retained a region if Bonferroni-corrected P-value > 8e−7) when compared to the average number of soft-masked bases in the autosomal sequences by calculating a z-score (54% of masked bases). Finally, we reduced the complexity of the contigs by overlapped the sequences with minimap2, converting the alignments into blast tabular format and detected the most likely unique sequences by a custom script. Briefly, we considered all alignments with >99% identity as referring to the same sequence, and only if each alignment spanned 95% of the total length of the shortest contigs involved. For example, an alignment of 296 bp with identity of 99.5% between contig1 (1000 bp) and contig2 (300 bp) would be considered, and only contig1 would be kept for downstream analyses.

Intersections between the different genomes have been visualised using the SuperExactTest package[59]. Motif enrichment was computed using HOMER (4.10.4)[60] on the non-reference sequences using all the genomes pooled together as background. Finally, sequences were characterized for gene content.

The proteins prediction was performed three ways: (1) using Augustus[61] (v.3.3.3) on the non-reference sequences with default parameters; (2) using Augustus (v3.3.3) on the sequences with 100 bp flanking regions included; and (3) aligning the sequences using DIAMOND (v2.0.6)[62] BLASTX to a database consisting of proteins from UniProtDB, SwissDB and 9 ruminants (taxa id 9845) RefSeq genomes downloaded from NCBI (GCF_000247795.1, GCF_000298355.1, GCF_000754665.1, GCF_001704415.1, GCF_002102435.1, GCF_002263795.1, GCF_002742125.1, GCF_003121395.1, GCF_003369695.1). Predicted proteins have been extracted through a custom python script and were aligned using DIAMOND[62] BLASTP to the same protein database previously described. We considered a high-confidence protein structure if the three methods consistently predicted the same complete protein structure, inclusive of start and stop sites.

The full pipeline, including the custom scripts used to generate all outputs, is accessible on GitHub (https://github.com/evotools/CattleGraphGenomePaper/tree/master/detectSequences)[63].

**Linear expanded genome**. Due to memory and computational constrains, we could not use the full mWGA to generate the set of vg indexes required to align and process short-read sequencing to a graph. Instead, we used autosomal chromosome-by-chromosome alignments of the five assemblies to generate a graph genome that can be successfully indexed with the vg[12] software allowing us to align reads and perform variant calling.

We generated a linear expanded genome with the purpose of providing an easy to use, expanded version of the cattle reference genome that is also easy to implement in current best practice pipelines. We extracted all nodes not present in the linear Hereford genome, but that were found in the other 4 assemblies considered using libbdsg (v0.3)[57]. Nodes were then labelled based on the genome in which they were found (i.e. a node can be from 1 to 4 different assemblies). The nodes were then trimmed for N-mers, and regions overlapping a candidate misassembled region in the N'Dama or Ankole genome were excluded. We then combined the regions if they were <5 bp apart using bedtools, and then labelled the regions depending on their proximity to a gap (<1000 bp from a gap) or to a telomere (10 Kb from the end of a chromosome or scaffold >5 Mb long), classified them based on their length (short if <10 bp, intermediate if between 10 and 60 bp and long if >60 bp) and whether they were haplotypes (<95% of the bases coming from a non-reference node) or novel (≥95% of the bases coming from a non-reference node). We retained all long regions (>60 bp), those not at telomeres and not flanking a gap. Finally, we excluded all regions that were too repetitive in comparison to the autosomes in the different genomes and sequences that were too similar, retaining only the largest of the two. For details of the selection of the NOVEL set of contigs, see section "Genome alignment and comparison" in Materials and Methods. This generated a final set of contigs that, once combined with ARS-UCD1.2, formed the final extended linear genome (ARS-UCD1.2+).

**Graph genome**. Comparatively few pieces of software capable of handling large genomes and graphs are currently available. Two in particular prove to be

particularly promising: the vg tools[12] and Seven Bridges graph genome pipelines[11]. In the current study we chose to apply the vg pipeline, which is able to call structural variants detected through multiple assembly comparisons. This is also supported by recent studies that have proven graph alignments to be superior in performance when alignments were generated through a reference-free comparison[64].

We first aligned the five cattle assemblies using CACTUS chromosome-by-chromosome (i.e. all chromosomes 1 from the five genome together). The CACTUS alignments were then converted to a vg graph using hal2vg (v2.1) (https://github.com/ComparativeGenomicsToolkit/hal2vg), dropping the ancestral genomes, referencing to the Hereford assembly and processed as recommended on the vg wiki page (VG5). We also generated second and third graphs with more and no diversity, respectively. To create the second graph, hereon called VG5p, we added >11 M short variants from 294 worldwide cattle[23] to the VG5 graph through the 'vg add' command. To create the third graph, we simply provided the linear ARS-UCD1.2 genome to 'vg construct' specifying the VCF with the 11 M variants described in Dutta et al.[23] (VG1p). To create the fourth and last graph, we simply provided the linear ARS-UCD1.2 genome to 'vg construct', without specifying any source of variation, and ultimately generating a graph representation of this single linear genome (VG1). The script used to generate the graphs are available on GitHub (https://github.com/evotools/CattleGraphGenomePaper)[63]. Finally, we used FORGe[31] to prioritize a subset of the 11 M variants for the three breeds considered (Angus, N'Dama and Sahiwal). We performed the prioritization using the hybrid method and selecting the top 10% of the variants considered.

We evaluated the performances of the graph genomes in two ways. We aligned to a variant-free linear graph based on the Hereford genome using vg (VG1). We also aligned and called variants using the standard BWA-HaplotypeCaller (bwa v 0.7.17; GATK v4.0.11.0)[65,66] and BWA-FreeBayes (FreeBayes v 1.3.1-16-g85d7bfc-dirty)[20] pipelines on the ARS-UCD1.2 genome.

All the graphs were generated using vg version 1.20.0. Short reads processing was performed using vg v1.22.0. Despite the change of version, the graphs generated in the version 1.20 can be used also in the next releases. All the script used for the analyses were generated through bagpipe (https://bitbucket.org/renzo_tale/bagpipe/src/master/).

Reads for the nine samples of three different breeds (Angus, Nigerian N'Dama and Pakistani Sahiwal) with a similar coverage (~30–50X) were considered for the analyses. Six of the nine samples were novel to this study with the three Angus taken from databases[67,68] (Supplementary Table 7). Whole blood for the three novel N'Dama samples was collected into PAXgene tubes, and DNA was extracted through the standard procedure as outlined in the PAXgene blood DNA kit handbook. Whole blood for the three novel Sahiwal samples was collected into EDTA tubes, and DNA was extracted through the standard procedure as outlined in the TIANamp Blood DNA Kithandbook (TIANGEN Biotech Co. Ltd, Beijing). Samples were then sequenced on a Illumina HiSeq X Ten at the Edinburgh Genomics sequencing facility. Samples were aligned using the guidelines reported in the vg GitHub wiki page, and implemented in the bagpipe pipeline (https://bitbucket.org/renzo_tale/bagpipe/src/master).

**Bionano optical mapping**. We generated ~100X OM data for two Kenyan N'Dama samples, one of which was an offspring of the assembled individual. Blood was collected by jugular venupuncture into EDTA vacutainers. Following erythrocyte lysis, monocytes were purified from the leukocytes using a positive selection MACS protocol with an anti-bovine SIRPα mono-clonal antibody (ILA-24, concentration 1 μg/ml[38]). Agarose plugs containing $5 \times 10^5$–$1 \times 10^6$ isolated monocytes were prepared using the Bionano Blood and cell culture DNA isolation kit (Bionano Genomics, San Diego, US) according to the manufacturer's instructions and the extracted DNA used for analysis on the Bionano Saphyr platform. Resulting reads were processed through the Bionano Solve pipeline (v3.3_10252018, refAligner v7915.7989rel). We then converted the resulting outputs to vcf through smap_to_vcf_v2.py. Then, we converted all non-translocation SVs into bed format expanding the initial and end positions defined by the Bionano Solve pipeline with the largest values defined by the confidence interval, and then added an additional kilobase to account for the resolution of OM data and uncertainty in the positions inherent in OM.

After generating bed intervals for each of the two individuals, we concatenated the bed files, sorted them, combined them through bedtools merge and, finally, retained the regions mapped on an autosomal region.

**Benchmarking the graph**. To evaluate the performances of the graph genomes we collected different metrics, which can be split into two categories: (a) read-based metrics and (b) variant-based metrics.

The first category includes the number of reads mapped to the genomes by the different algorithms, and how many of the reads called by vg are perfectly mapped.

The second category includes metrics based on the variants called, including number of variants identified, DP, transitions/transversions rate and allelic balance (i.e. the ratio of reads supporting the reference and the alternate allele used for the variant calling). These metrics have been computed for different variant lengths to see how the callers perform with different types of variants, using the script available on GitHub (https://github.com/evotools/CattleGraphGenomePaper)[63]. The analyses have been carried out considering (a) the variants present in the given

graph as known and all other as novel, and (b) the 11 M variants as the set of known variants and all the other as novel.

After gathering overall metrics, we focused our attention on large structural variants called by vg on the VG5p graph, since these are the hardest to genotype with current broadly adopted methods. First, we combined variants across the nine samples using bcftools (v1.10) merge, and checked how many overlapped with OM signals detected on two N'Dama samples. Although being called for two different samples than the N'Dama sequenced, it can still provide insights into N'Dama-shared variants not present in the current linear genome. We assessed the significance of the overlap by randomly selecting 10,000 times regions of the same sizes as the detected ones and overlapping them with the OM data to estimate a Z-score. We defined the size of a structural variant as equal to the size of the reference allele. Also, we checked whether the size distribution of indels in genes shows a higher number of in-frame than out-of-frame variants (i.e. insertions and deletions of size multiple of 3 versus rest). Second, we checked if the structural variants called for the different breeds overlapped differently with the OM data to assess whether individuals genetically closer to the two N'Dama genotyped with OM have a proportionally higher number of overlaps between graph-based and OM structural variants.

Third, we investigated high-quality, group-specific large structural variants identified by vg. We iteratively intersected individuals of a target breed with samples of the other two breeds using bcftools isec, retaining a variant if found only in the target individual (e.g. we intersect Angus1 with Sahiwal1; then, we keep the specific variants for Angus1, and intersect it with Sahiwal2, and so on). Then, samples of the same breed are combined with bcftools merge, that kept all variants found in at least one animal of the same breed. Then, we retained a variant if they had high quality (QUAL > 30), DP close to the expected value ($20 < DP < 90$) and allowing no missingness and with sufficient evidence for the alternate allele (non-reference allele count ≥ 5). Finally, we focused on variants with length > 500 bp in order to keep the results comparable with the OM and allowing direct comparison with the N'Dama samples.

We compared the structural variants from the graph with the ones called from Delly2 (v0.8.5)[35]. Variants called by Delly2 for each individual with no soft-filter and high quality (QUAL > 30) were retained. Individuals' SVs of the same type were combined using SURVIVOR[69] (v1.0.7), allowing 100 bp of distance between break points, not accounting for the strand, retaining only SV longer than 500 bp and excluding translocations. These were then intersected with the OM regions. We also combined the samples of the same breed as done for the graph genome, retaining variants with no missingness and sufficient support for the alternative allele (non-reference allele count > 5), dropped translocations and finally, intersected with the regions from the OM analysis.

Finally, we compared SVs called from Delly and VG5p based on their type (insertions, deletions, inversions and duplications). This approach, though more consistent, comes with limitations since the different callers call different types of SV: VG5p can only call insertions, deletions and complex SV, with the latter inclusive of inversions and more complicated rearrangements (e.g. a substitution and a deletion at the same site); Delly can call only precise deletions, duplications and inversions; finally, the OM can call insertions, deletions, inversions and duplications. SVs called from VG5p were first broken into single-allele variants using vcfbreakmulti from vcflib (v1.0.1)[70] annotated using vcf-annotate --fill-type from the vcftools library[71]; the variants were then split by annotated type, multiallelic SV recombined with vcfcreatemulti and converted to BED format using SnpSift[72] (v 4.3t build 2017-11-24 10:18) and a series of custom scripts. Delly variants were separated based on the alternate allele field into separate SVs, and similarly SVs from OM were split by the SVTYPE annotated field. Insertions and deletions from VG5p were then intersected using bedtools (v2.30.0) with insertions and deletions from OM, respectively. Analogously, deletions, duplications and inversions from Delly were intersected with the same categories from OM data using bedtools (v2.30.0). Resulting unique SVs were combined and counted as number of consistent, overlapping SV.

**ATAC-seq data processing**. Illumina paired-end reads for B-cells of three samples (1 Holstein-Friesian, 1 N'Dama and 1 Nelore) were generated using Illumina HiSeq X Ten at the Edinburgh Genomics facility. Details on the preparation of the DNA libraries can be found in Supplementary Methods 1. In addition to the three samples, one nucleosome-free DNA sample was processed to identify and exclude false positives. All read accession numbers are listed in Supplementary Table 7.

We processed paired-end reads as follow: we first trimmed the reads, extracting only the paired ones with length ≥ 36 bp using trim_galore (v0.6.3)[73]. As a spike-in of mouse cells had been used in these samples trimmed reads were aligned to the target genome concatenated with the mouse genome GRCm38 using bowtie2 (v2.3.1) and only one mapping per read was saved in order to account for repetitive elements (parameters -X 1000 --very-sensitive). Reads aligned to the mouse genome and mitogenome were excluded with samtools and peaks were called using Genrich (v 0.5_dev, parameters: -j -r -e MT -v). The full pipeline to process the samples was generated using bagpipe (https://bitbucket.org/renzo_tale/bagpipe/src/master). We also compared the effect of using only uniquely mapped reads when peak calling. We aligned the reads as previously described to ARS-UCD1.2 and ARS-UCD1.2+, and then retained only reads uniquely mapped using Sambamba

(v0.5.9; command view -h -f sam -F "[XS] = = null and not unmapped and not duplicate").

We called peaks on all five linear assemblies and ARS-UCD1.2+ separately. For each sample, we excluded peaks overlapping a peak in the nucleosome-free DNA sample for more than 50% of their length (bedtools subtract -A -f 0.5), which were considered as false positive peaks. We then calculated the Q-scores for each peak using the Benjamini-Hochberg correction, setting the number of independent tests to the theoretical size of the cattle genome (2.7 Gb). For each region, we also checked which one did not overlap a masked region in the respective assembly for at least 40% of its length.

Heatmaps have been created using Deeptools (v3.5.1)[74] with the aligned reads as inputs, first filtering out reads mapping to the mouse spike-in genome and then converting them to bigWig using bamCoverage (options --minFragmentLength 35 --maxFragmentLength 150 --normalizeUsing RPGC -bs 10 -e --effectiveGenomeSize 2779691414). The generated bigWig files are then used as inputs to computeMatrix (reference-point mode with parameters -a 3000 -b 3000 --missingDataAsZero --skipZeros) using the ARS-UCD1.2 annotation (Ensembl version 103) and the genes predicted by Augustus as annotations.

**Reporting summary**. Further information on research design is available in the Nature Research Reporting Summary linked to this article.

## Data availability
The long reads and short read data for the Ankole assembly have been deposited in the ENA database under project accession code (PRJEB39282). The long read and short reads data for the N'Dama sample have been deposited in the ENA database under project accessions codes (PRJEB39330) and (PRJEB39334). The short read sequencing for the three Sahiwal and the three N'Dama samples have been deposited in the ENA database under project accessions codes (PRJEB39352) and (PRJEB39353), respectively. The N'Dama (GCA_905123515) and Ankole (GCA_905123885) assemblies have been deposited in the ENA database under accession codes (PRJEB41519) and (PRJEB41564), respectively. The optical mapping reads for the two N'Dama samples have been deposited in the ENA database under accession code (PRJEB47998). The ATAC-seq reads have been deposited in the ENA database under accession code (PRJEB49075). Output for the analyses can be visualised in (BOmA)[www.bomabrowser.com/cattle]. Source data are provided with this paper. Finally, the five VG graphs (VG1, VG1f, VG1p, VG5 and VG5p) as well as the CACTUS five-way whole genome alignments have been uploaded on Zenodo with doi 10.5281/zenodo.5749842 and 10.5281/zenodo.5750390. Source data for Figs. 1, 4B–D, 6A and Supplementary Fig. 1 are provided with this paper. Source data are provided with this paper.

## Code availability
Code for running the analyses presented in this work can be found at https://github.com/evotools/CattleGraphGenomePaper and on Zenodo with https://doi.org/10.5281/zenodo.5749432[63].

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

## Acknowledgements

The study was funded by grants BB/T019468/1, BB/R015155/1, BB/P024025/1, BBS/E/D/10002070 and 5682306 from the BBSRC to JGDP. Authors are grateful to Dr. Maryam Muhammad and Dr. Edward Amali for assisting with the collection of the N'Dama samples involved in this study. We thank Dr. Maria Eugenia Z. Mercadante for supplying samples of Nelores from the herd of the Centro Avançado de Pesquisa Tecnológica do Agronegócio de Bovinos de Corte, Sertãozinho, SP, Brazil and are very grateful for the help of Professor Isabel Santos of the University of São Paulo for the assistance in processing the samples. This research was also funded in part by the Bill & Melinda Gates Foundation and with UK aid from the UK Foreign, Commonwealth and Development Office (Grant Agreement OPP1127286) under the auspices of the Centre for Tropical Livestock Genetics and Health (CTLGH), established jointly by the University of Edinburgh, SRUC (Scotland's Rural College), and the International Livestock Research Institute. The findings and conclusions contained within are those of the authors and do not necessarily reflect positions or policies of the Bill & Melinda Gates Foundation nor the UK Government.

## Author contributions

J.G.D.P. conceived the study, A. Talenti and J.G.D.P. designed the analyses and A. Talenti and J.G.D.P. performed them. L.J.M., T.C. and P.W. contributed to the conceptualization of the study. S.J. developed and managed the BomA browser. J.G.D.P. and A. Talenti prepared the initial manuscript with all authors contributing to subsequent drafts. J. Powell generated the chromatin data. E.P., J.D.H., J. Powell, E.T.O. and T.C. prepared the DNA and Bionano samples. D.W., P.T., W.A., D.M., E.A.J.C., C.E., E.T.O., E.R.A., A. Tijjani, K.M., A.F., B.R.F., A.Q., U.C. and P.W. provided samples and expertise for the studies.

## Competing interests

The authors declare no competing interests.
