## [Peer Review File · Nature Communications]

REVIEWER COMMENTS

Reviewer #1 (Remarks to the Author):

Summary: In this manuscript, Talenti et al. present two new long-read based reference genome assemblies of the N'Dama and Ankole cattle breeds. These two breeds are of African-origin, and are acknowledged in the literature to be diverse from the European and Indicine breeds of cattle due to extensive admixture events among all three. These two new assemblies are novel and could be of use to the community, but the methods that the authors document to generate the assemblies are quite convoluted due to a paucity of X coverage in long reads. The novelty of sequence reported in this study could also be of use to the community, but it appears that most of the novel sequence resides within repetitive elements. Further analysis and validation are required to ensure that the statistics reported are accurate and relevant to the research community. I have separated my concerns into Major comments and Minor comments, and listed them in the order in which I encountered them in the text.

Major comments:

Novel sequence analysis:

Line 151: There are several problems in this comparison that must be addressed with further analysis and explanation. Firstly, the authors acknowledge in their methods that the sex chromosomes of the Ankole and N'Dama assemblies they created were extraordinarily difficult to assemble. If the majority of unique sequence is from fragmentary contigs that originated on the X chromosome, would those not be difficult to polish in the final assemblies? Aren't these unplaced contigs also difficult to align in an MSA due to the fact that they are often heterozygous regions of autosomal scaffolds? Also, the high proportion of repetitive content (~73-93% of reported novel sequence) is difficult to validate as truly "novel" sequence in the assembly. Isn't it more likely that these repetitive novel regions are a result of (A) misassembly through repetitive element compression/expansion (CE), (B) mobile element insertion (MEI) events or (C) misalignment in the MSA? I believe that read alignment statistics can be used to test these possibilities in each case. Tools such as FRC_align can be used to identify CE events to filter (A), and there are several paired-end SV callers that can identify candidate MEI that are novel to each assembly. Detecting (C) is more difficult, but may have an acceptable error rate after the aforementioned steps are performed.

ATAC-Seq peak analysis:

Line 296: I do not understand what this analysis provides to the overall narrative of the manuscript. Nor do I agree that an enrichment of peaks is an appropriate measure to judge the "utility" of an assembly here. Finally, the use of a comprehensive linear reference containing the superset of all unique sequence of all assembled cattle is expected to have more regions that will appear as peaks. I would be more convinced if the novel peaks discovered in the ARS-UCD1.2+ reference were found to be from uniquely mapped reads and not artifacts from the recruitment of reads that mapped to alternative locations in the base ARS-UCD1.2 reference. Without this confirmatory analysis, I believe that this section could be removed from the manuscript as it does not add additional information.

Assembly methods:

In general: The use of PacBio reads less than 50X coverage to generate both assemblies may have been the major limitation of this study. Particularly for extremely outbred individuals (as can be expected for N'Dama and Ankole), lower X coverage can often result in extremely fragmented genome assemblies. The extensive computational analysis documented in the supplementary files notwithstanding, I have reservations regarding the accuracy of novel assembled sequence and scaffolding in this study, particularly with the N'Dama assembly (which did not use Optical mapping data for scaffolding).

Line 425: I am quite concerned with potential bias that may have been incorporated into the N'Dama assembly through the use of comparative alignment scaffolding. How were contigs determined to be chimeric? Only through comparative assembly alignments? Would this not mask unique N'Dama structural variants if not verified with orthogonal data? Also, the assembly of the N'Dama X chromosome is poorly described in the statement, "[t]he unplaced fragments have then been used to assembly the sex chromosomes." How was the N'Dama X chromosome assembled and why was it so fragmented as to warrant extended detail in the methods section? Metrics on the number of contigs and a rationale for their incorporation into "pseudo-X" chromosome scaffolds are needed here.

Line 455: Supplementary document 1 details methods not reported in the main text, including the use of the Canu assembler to generate the N'Dama assembly. I am not opposed to presenting all methods used in an analysis; however, I think that further curation of these details is needed. I was very confused by the layout of the document and felt that it needs extensive reorganization and it must cite all figures to the appropriate sources. Several figures were taken without citation from the Canu and Falcon-Unzip websites or publications and this must be corrected as soon as possible. I have noticed that this is a problem in the other supplementary documents as well.

Line 456: Although quickmerge can be used to merge long-read assemblies, it was primarily developed to merge assemblies constructed from short-reads and long-reads. As such, the quickmerge documentation requests that the assemblies to be merged should first be checked for misassemblies and split into smaller constitutive contigs. I note that this is not documented in supplementary document 2, so I am assuming that this was not done prior to the quickmerge step. I really do not see the reason to perform quickmerge here and suspect that it may have actually incorporated errors into the final assembly. If the authors used BioNano scaffolding, why was that not used on the best assembly to avoid further errors? Also, the authors do not document the amount of chimeric errors fixed by the BioNano "solve" pipeline.

Comparison of graph genome resources:

Line 538: I am unsure why the authors did not create a VG1p variant as well. In the introduction, they reference the use of graph-based variant calling using only variant information, but they do not test the utility of variant calls distinct from their 5-way multiple sequence alignment graph. Also, the current version of the "vg" tool does not appear to have an "add" command (or at least one that is listed in the "readme"). Did the authors use the "augment" command here? Were reads aligned to the graph before variants were added?

Minor comments:

Line 126: A citation is needed to support claims that the published assemblies are a poor representation of global diversity of cattle breeds. I don't necessarily disagree with the statement, but precision (ie. how representative the assemblies actually are) is needed here.

Line 128: It would be more accurate to state that the PacBio long reads were used in the assembly and the Illumina paired end reads were used to polish the assemblies.

Line 142: I do not think that Figure 1 demonstrates the quality of the assemblies. Instead, I would be more convinced by data including statistics such as feature response curves as calculated by the FRC_align pipeline and read alignment mapping histograms. Figure 1 just shows an ordination plot of the short-read data separated by breed of cattle.

Figure 2: The caption should be updated to state that the sequence in the upset plot (A) comparison is based on the supposed "unique" sequence in each assembly. This was not clear from the original caption.

Figure 4: the caption and figure legend should make it clearer that comparisons of alignment algorithms are between the "vg" tool (VG5 and VG5p) and BWA MEM (VG1). This is a major caveat to this comparison that should be listed first and foremost.

Line 401: Was an approved institutional animal care and use protocol followed for tissue sampling?

Line 446: What polishing software was used for the N'Dama assembly?

Line 449: There is an error statement in the text here.

Line 542: The link to the Github repository is broken.

Line 593: The link to the Github repository is broken here as well.

Reviewer #2 (Remarks to the Author):

The paper by Talenti and colleagues presents a bovine graph genome that integrates the current Hereford-based reference sequence and structural and single nucleotide variants detected from different breeds of taurine and indicine cattle. It is increasingly recognized that the use of a linear reference genome does not appropriately capture genetic diversity within a species. Talenti and colleagues augment the taurine reference genome with variants detected in indicine cattle to make the reference sequence more informative for cattle from "underrepresented" breeds. They detect a total of 202 Mb non-reference sequences.

The paper presents four different vignettes. The assembly of two African cattle genomes is followed by the construction of a multi-assembly graph, subsequently, variant calling from an augmented reference structure is presented, followed by an analysis of ATACseq peak calling from an augmented reference.

In the first vignette, the authors assemble the genomes of two African cattle using PacBio CLR reads. Although the availability of genome assemblies for African cattle is a valuable resource, the lack of description of these assemblies dampens enthusiasm. Scaffolding was done using either the "previously published genomes" (which ones???) or Bionano data. The quality of the assemblies is reasonable as indicated by N50 values and BUSCO scores. However, the assemblies are less contiguous than those recently generated by trio binning or the use of HiFi data. The supporting files describing the assembly process are merely a sequence of shell scripts and should be revised to present the process in a readable way (or at least a synopsis should be presented). More details about the assembly need to be presented in the results section, e.g., the sex of the animals, the number of contigs and scaffolds, as well as the assembly length and quality. Moreover, according to the discussion, the assemblies were created from cattle of African taurine and taurine x indicine ancestries. This should be presented in the results section. The ancestries of the two assemblies need to be related to the findings presented in Figure 2.

In the second vignette, the CACTUS aligner is applied to detect sequences missing in the current bovine reference genome. The paper reports 202 Mb non-reference sequences from two novel African cattle assemblies and two previously generated genome assemblies from a Angus x Brahman cross (<https://doi.org/10.1038/s41467-020-15848-y>). The way the results are presented is confusing, particularly Figure 2, and needs a thorough revision. Are the units correct (1e8)? My interpretation of Figure 2 is that the five horizontal bars represent the sizes of the assemblies and the vertical bars represent the amount of sequences that are shared between the assemblies. What is the sum of all intersections presented in this Figure? Are the values supposed to sum up to 202 Mb or to 2995 Mb (Supplementary Table 3; why have some numbers in Table S3 digits?), or to a totally different value.

From my understanding, Figure 2 indicates a problem with the Angus assembly. According to data curated at NCBI (GCA_003369695.2), the Angus assembly is only slightly smaller than the Brahman assembly (2580 vs 2680 Mb), and the difference is even lower, when considering only autosomal sequences (2468 vs 2478 Mb). A high vertical bar indicates a large amount of sequences is present in all but the Angus assembly. This is weird, because it is clearly too small to be attributable to the X chromosome (~140 Mb). Also, the horizontal bar indicates the size of the Angus assembly is less than half of Hereford, which seems not reasonable. As one of the African cattle genomes originates from a taurine x indicine cross, I'm wondering if this assembly contributes more non-reference sequences than the other.

In the third vignette, the authors augment the Hereford-based reference with non-reference sequences to create a graph genome. Compared to a recently published analysis of a bovine genome graph (<https://doi.org/10.1186/s13059-020-02105-0>), the results shown here are presented superficially and don't provide novel insight. It is unclear how the graph was created (methods: "we aligned the genomes chromosome by chromosome..."). Is this approach different to the CACTUS-approach applied earlier? Did the graph contain the 202 Mb non-reference sequences that were detected previously, or only a subset of these? Did the graph contain only autosomal sequences, or also unplaced contigs, and sex chromosomes? The way VG5p was created from the 11 M short variants remains elusive. Previous studies showed that variants need to be prioritized before adding them to a graph (<https://doi.org/10.1186/s13059-018-1595-x>). Also, a haplotype index may be considered to add only biologically plausible paths in the graph. The comparison between VG and BWA is not really appropriate, because VG may provide better mapping accuracy from an empty graph than BWA. Thus, findings need to be related to VG1. The paper shows that VG5p did not really perform better than VG5 (Figure 4), possibly because the variants were not prioritized accordingly. Would VG5 enable better mapping accuracy than a graph constructed from only 11M variants (no structural variants added) that were appropriately prioritized?

The fourth vignette deals with ATACseq peak calling. It seems that this analysis is largely disconnected from the previous sections, because it relies on a linear reference sequence that has been expanded with non-reference sequences. A similar approach was presented recently for ChIPseq calls (<https://doi.org/10.1186/s13059-020-02038-8>)

The discussion somewhat fails to put the work into context as it neglects a broad body of literature on recently created whole-genome graphs.

Reviewer #3 (Remarks to the Author):

Talenti et al. - A cattle graph genome incorporating global breed diversity

The authors have created pan-genomes based on reference assemblies from 5 different cattle breeds, and showed the benefits of such a graph genome. They assessed variant calling including larger structural variation, and ATAC-seq benefits. Showing the benefits of graph genomes for functional analysis. It is a very extensive and well conducted study in a rather new field of interest to the community.

General comments

I believe the introduction should not include the breeding part (L77-81), but keep focus on the actual benefits of a graph genome. For breeding a graph genome is not directly needed (although it may

benefit from findings). I guess the main benefit of graph genomes are the detection of structural variants and genetic differences that are breed specific, and hence could aid detection of causal variants for e.g. disease resistance. So the advantage is really on the functional detection directly, and on breeding indirectly after discovery of functionally important variants, but only if a breeding program including genomic selection is in place, which would be the major challenge in African countries.

Is a pan-genome better than a breed (or line) specific reference genome? In order to create a pan-genome, you need to make a de novo assembly of other breeds, in the future many breeds will have their own reference genome to align against, will a pan-genome be needed then? I believe that SV calling is also improved when using a breed specific ref genome. The comparison of for instance N'Dama reference vs VG5 has not been made (or I missed it), and would be valuable to add. I think the pro's and con's of a breed specific reference genome versus a graph genome should at least be discussed.

For the variant calling freebayes and GATK were applied to ARS1.2 genome, hence they can be directly compared to VG1. Looking at the results in Supplementary document 3 VG1 performs poorer than the commonly used GATK (but better than freebayes). Please also comment on that. Any hints on why this happens? More importantly what does it mean for using graph genomes in general? You focus on the allelic balance but there are other features, e.g. transition-transversion drop for unknown angus variants for VG5, variant quality for VG1 low

The results section can benefit from some sub-sub headings to aid the reader when switching topic within a sub-section.

The links to the github are not working because the folder is not there yet, please make sure it is available before publishing.

Line by line comments

L61 I think the wording should be changed a bit, as the 1000 bull genomes project actually also facilitates indicus breeds (see presentation on their website), does variant calling for indicus specific variants as well as a combined indicus and taurus set. Yes there are far more European breeds in their resource, but that has to do with more funding being available for those breeds.

L80-81 Do you really believe a reference population augmented with local breeds is the main factor to drive selection in Africa? There is an indicus HD SNP array that will capture genomic relationships to circumvent the lack of accurate pedigree recording. Accurate trait recording would be essential to start a good breeding program. To organise the structure and centralise it, that is the main difficulty for breeding.

I guess an pan-genome including African breeds would be useful to identify causal variants for e.g. disease resistance as mentioned later on. Please see my general comments to improve the introduction in this respect.

L199 Change 'with 163M and 194M nodes and edges' into 'with 163M nodes and 194M edges'

L240-242 Freebayes is not designed to call large variants (not larger than the read length). For GATK haplotypcaller also only SNP and indels can be called. So not fair to compare. However, it is worthwhile to mention that VG can call such large variants together with short variants. Just wondering how accurate they are. To properly address vg accuracy for calling large variants you will need to compare the results to some SV callers that can call variants of similar size and type.

L252 Change NDama into N'Dama

L274-275 Wasn't it the VG5p graph that was used for the SV calling (with vg), rather than the Hereford (VG1), as indicated in line 246

L352-354 Is it just insertions and deletions that can be called or also more complex SV like inversions?

L416 Please define OM

L449-450 Please fix reference error: Error! Reference source not found

L506+542 The github links give errors, going to github.com/evotools shows that the folder for this

paper is not there

L558-560 Sentence 'Samples were then sequenced on a Illumina HiSeq X Ten at the Edinburgh Genomics sequencing facility.' Repeated, please remove one.

L617 With this one heterozygote allowed it is not clear to me what you mean. Does at least one individual have to be heterozygote? What if the only individual carrying the variant is homozygous (if that even occurred). I guess you want to be certain that you have at least one carrier with sufficient evidence hence the minimum allele count of 5. I guess if my reasoning is correct you can simply add 'at least one heterozygote with sufficient evidence for the alternative allele'.

Figure 2A It took me a while to understand the figure. Perhaps a table would be easier to understand

Figure 5A What do the greyish rectangles indicate in between?

Supplementary material

- Please add table & figure descriptions to explain the viewer what is in there.
- Figures do not always fit on a page, please adjust.
- In table 6 under nSeq is that the number of contigs? If so please change into the more informative 'nContig'
- In table 10 the final bargraph does not include a bar for ARS+ which is indicated in the legend

RESPONSE TO REVIEWER COMMENTS

We would like to express our sincere gratitude to all three reviewers whose inciteful and helpful comments have helped to substantially improve the quality of the manuscript and analyses. Changes to the manuscript include, but are not limited to:

- Improving the approach used to define high-quality novel sequence absent from the Hereford genome. Within which we now define novel putative genes that we show are enriched with open chromatin at their predicted transcription start sites consistent with functional regions.
- Adding comparisons to a new graph, VG1p, as suggested i.e. the Hereford genome augmented with 11 million variants. We illustrate how this does not lead to the same improvements observed for the graphs incorporating all five assemblies (VG5 and VG5p)
- We have now also added comparisons to structural variants calls using Delly v2, a traditional SV caller. We show that in comparisons to the independent optical mapping data the vg SV calls are more consistent.

These along with the further changes and improvements we outline in greater detail below we believe have substantially strengthened the manuscript.

Reviewer #1

Summary: In this manuscript, Talenti et al. present two new long-read based reference genome assemblies of the N'Dama and Ankole cattle breeds. These two breeds are of African-origin, and are acknowledged in the literature to be diverse from the European and Indicine breeds of cattle due to extensive admixture events among all three. These two new assemblies are novel and could be of use to the community, but the methods that the authors document to generate the assemblies are quite convoluted due to a paucity of X coverage in long reads. The novelty of sequence reported in this study could also be of use to the community, but it appears that most of the novel sequence resides within repetitive elements. Further analysis and validation are required to ensure that the statistics reported are accurate and relevant to the research community. I have separated my concerns into Major comments and Minor comments, and listed them in the order in which I encountered them in the text.

Major comments:

Novel sequence analysis: Line 151: There are several problems in this comparison that must be addressed with further analysis and explanation. Firstly, the authors acknowledge in their methods that the sex chromosomes of the Ankole and N'Dama assemblies they created were extraordinarily difficult to assemble. If the majority of unique sequence is from fragmentary contigs that originated on the X chromosome, would those not be difficult to polish in the final assemblies? Aren't these unplaced contigs also difficult to align in an MSA due to the fact that they are often heterozygous regions of autosomal scaffolds?

Also, the high proportion of repetitive content (~73-93% of reported novel sequence) is difficult to validate as truly "novel" sequence in the assembly. Isn't it more likely that these repetitive novel regions are a result of (A) misassembly through repetitive element compression/expansion (CE), (B) mobile element insertion (MEI) events or (C) misalignment in the MSA? I believe that read alignment statistics can be used to test these possibilities in each case. Tools such as FRC_align can be used to identify CE events to filter (A), and there are several paired-end SV callers that can identify

candidate MEI that are novel to each assembly. Detecting (C) is more difficult, but may have an acceptable error rate after the aforementioned steps are performed.

We thank the reviewer for their comments. We have now extensively changed the methods for detecting novel sequences in the different genomes to address the concerns of the reviewer. In particular, we identified the novel portions of the genome from the nodes in the graph, which allows us to have a more precise definition of what is non-reference, therefore reducing the risk of misalignments (issue C). In the process, we then excluded all nodes falling into a CE feature predicted by FRC_Align using short reads sequencing on our two *de novo* genomes, reducing the risk of CE (A). Furthermore, we now characterise novel segments that were not significantly more repetitive than the average of the autosomal sequences (Z-scores calculated on the average number of masked bases in each assembly) reducing the risk of MEI (B). We believe that these changes make the identification of the novel sequences from the non-Hereford genomes more reliable and robust. More details on this revised approach can be found at lines 152-181.

ATAC-Seq peak analysis: Line 296: I do not understand what this analysis provides to the overall narrative of the manuscript. Nor do I agree that an enrichment of peaks is an appropriate measure to judge the “utility” of an assembly here. Finally, the use of a comprehensive linear reference containing the superset of all unique sequence of all assembled cattle is expected to have more regions that will appear as peaks. I would be more convinced if the novel peaks discovered in the ARS-UCD1.2+ reference were found to be from uniquely mapped reads and not artifacts from the recruitment of reads that mapped to alternative locations in the base ARS-UCD1.2 reference. Without this confirmatory analysis, I believe that this section could be removed from the manuscript as it does not add additional information.

We thank the reviewer for the suggestion. We expanded the analysis also including the reads uniquely mapped to the ARS-UCD1.2 and ARS-UCD1.2+ genomes, showing that the latter calls a larger number of peaks compared to the former, and that the increase is above that expected simply due to the increase of the genome length (we called at least 1.5% more peaks despite an increase in genome length of only 0.7%). Moreover, we extensively changed the ATAC-seq part, using the smaller subset of novel contigs with a fraction of repetitive elements not significantly higher than the rest of the genome. We then expanded the analysis by performing a gene prediction on these sequences, detecting which are potentially functional, and intersecting these with the ATAC-seq peaks to demonstrate how novel genes and functional regions appear to reside in these novel sequences. See lines 372-382 and 395-405 for more details.

Assembly methods: In general: The use of PacBio reads less than 50X coverage to generate both assemblies may have been the major limitation of this study. Particularly for extremely outbred individuals (as can be expected for N'Dama and Ankole), lower X coverage can often result in extremely fragmented genome assemblies. The extensive computational analysis documented in the supplementary files notwithstanding, I have reservations regarding the accuracy of novel assembled sequence and scaffolding in this study, particularly with the N'Dama assembly (which did not use Optical mapping data for scaffolding).

To further address these concerns from the reviewer we added additional metrics in the main manuscript at lines 143 to 147 and supplementary document 1, including QV calculated through Merqury, coverage plots and FRC_Align analyses. These analyses confirm that although the sexual chromosomes are, as expected, fragmented (as is the case for other assemblies), the autosomal sequence is of good quality (QV>30) and coverage.

Line 425: I am quite concerned with potential bias that may have been incorporated into the N'Dama assembly through the use of comparative alignment scaffolding. How were contigs determined to be chimeric? Only through comparative assembly alignments? Would this not mask unique N'Dama structural variants if not verified with orthogonal data? Also, the assembly of the N'Dama X chromosome is poorly described in the statement, "[t]he unplaced fragments have then been used to assembly the sex chromosomes." How was the N'Dama X chromosome assembled and why was it so fragmented as to warrant extended detail in the methods section? Metrics on the number of contigs and a rationale for their incorporation into "pseudo-X" chromosome scaffolds are needed here.

The chimeric contigs have been detected and corrected using Ragout2 and alignments to the three other assemblies. Although it cannot be excluded that there are some remaining errors in any assembly, Ragout2 can detect and fix misassemblies and chimeras. In the original Ragout2 manuscript they highlighted that chromosome assembly with Ragout2 led to a similarly small number of structural errors as assemblies using bionano and HiC data (see <https://genome.cshlp.org/content/28/11/1720.full>). Moreover we show later in the manuscript good agreement between NDama optical mapping data and SV calls from the graph genome. Concerning the pseudo-X, we are aware of the limitations of assembling sexual chromosomes especially at lower coverage. We performed a reference assisted scaffolding using the reference chromosomes from the Hereford, Brahman and Angus genomes, which it should be noted are themselves mostly incomplete (trio-binning only provides one sex chromosome per breed and the Angus Y chromosome is only about 15Mb long). However, the scaffolding of the X/Y happened only after the scaffolding of the autosomes, limiting the types of contigs that can be mistakenly placed into these chromosomes. We've simplified and expanded the description of the scaffolding in Supplementary Document 1. We hope that the edited Supplementary Documents will make the rationale behind the scaffolding clearer.

Line 455: Supplementary document 1 details methods not reported in the main text, including the use of the Canu assembler to generate the N'Dama assembly. I am not opposed to presenting all methods used in an analysis; however, I think that further curation of these details is needed. I was very confused by the layout of the document and felt that it needs extensive reorganization and it must cite all figures to the appropriate sources. Several figures were taken without citation from the Canu and Falcon-Unzip websites or publications and this must be corrected as soon as possible. I have noticed that this is a problem in the other supplementary documents as well.

Thanks. We have now performed an extensive rewriting of the results, materials and methods and supplementary materials to better describe the method applied.

Line 456: Although quickmerge can be used to merge long-read assemblies, it was primarily developed to merge assemblies constructed from short-reads and long-reads. As such, the quickmerge documentation requests that the assemblies to be merged should first be checked for misassemblies and split into smaller constitutive contigs. I note that this is not documented in supplementary document 2, so I am assuming that this was not done prior to the quickmerge step. I really do not see the reason to perform quickmerge here and suspect that it may have actually incorporated errors into the final assembly. If the authors used BioNano scaffolding, why was that not used on the best assembly to avoid further errors? Also, the authors do not document the amount of chimeric errors fixed by the BioNano "solve" pipeline.

We had run quickmerge using very stringent cut-offs for combining contigs (see Supplementary document 2), followed by fixing any misassemblies using the optical mapping data. As can be seen in supplementary document 2 the use of quickmerge in this way substantially improved the contiguity of the assembly, and subsequent validation with the optical mapping data did not suggest

this was due to false contig joining. Although the optical mapping data did indeed correct 956 conflicts in 65 contigs, roughly half of them were contigs <1Mb, that were therefore very unlikely to have been combined by quickmerge. Several larger contigs were fragmented into 2-4 fragments. About 9,000 contigs were not cut from the scaffolding stage, including several very large contigs. We have extensively rewritten the supplementary document 2, detailing these steps in greater detail.

Comparison of graph genome resources: Line 538: I am unsure why the authors did not create a VG1p variant as well. In the introduction, they reference the use of graph-based variant calling using only variant information, but they do not test the utility of variant calls distinct from their 5-way multiple sequence alignment graph. Also, the current version of the “vg” tool does not appear to have an “add” command (or at least one that is listed in the “readme”). Did the authors use the “augment” command here? Were reads aligned to the graph before variants were added?

Thanks for the suggestion, we have now included a VG1p graph in the analyses (starting from lines 209). The results from this graph show how the 11M variants improved the number of mapped reads and of perfectly aligned reads, and to some extent also the allelic balance for single nucleotide polymorphisms. However, this version of the graph confirms that incorporating the multiple assemblies into the VG5 and VG5p graphs substantially improves SV calling. The command used was vg add. The reviewer is right that the command is not shown in the README, but it can be seen in the help output of the software at runtime.

Minor comments:

Line 126: A citation is needed to support claims that the published assemblies are a poor representation of global diversity of cattle breeds. I don't necessarily disagree with the statement, but precision (ie. how representative the assemblies actually are) is needed here.

We have added some references to the text as requested to supplement the PCAs shown in Figure 1 (see lines 69-76).

Line 128: It would be more accurate to state that the PacBio long reads were used in the assembly and the Illumina paired end reads were used to polish the assemblies.

Thanks. We have now changed the text accordingly (lines 124-128).

Line 142: I do not think that Figure 1 demonstrates the quality of the assemblies. Instead, I would be more convinced by data including statistics such as feature response curves as calculated by the FRC_align pipeline and read alignment mapping histograms. Figure 1 just shows an ordination plot of the short-read data separated by breed of cattle.

We thank the reviewer for the suggestion. We have now added several novel analyses to supplementary documents 1 and 2, including those from FRC_align as suggested, as well as a new figure (Figure 2) summarising the key metrics for the two genomes.

Figure 2: The caption should be updated to state that the sequence in the upset plot (A) comparison is based on the supposed “unique” sequence in each assembly. This was not clear from the original caption.

As discussed above we have now changed the method used to identify the novel portions of the genome to make it reference-based. This led to generating a new figure that shows the non-(Hereford) reference portions of the genome (now Figure 3). Thanks, we believe these new analyses motivated by the reviewer's comments have improved the quality of the final set of contigs.

Figure 4: the caption and figure legend should make it clearer that comparisons of alignment algorithms are between the "vg" tool (VG5 and VG5p) and BWA MEM (VG1). This is a major caveat to this comparison that should be listed first and foremost.

Thanks. We have now clarified the caption.

Line 401: Was an approved institutional animal care and use protocol followed for tissue sampling?

Yes, all protocols involving animals were approved prior to sampling by the relevant institutional animal care and use committee (ILRI IACUC or Roslin Institute Animal Welfare Ethical Review Body). All blood sampling was carried out by trained veterinarians, according to the approved institutional protocols. We now highlight this at lines 514-517.

Line 446: What polishing software was used for the N'Dama assembly?

We have now added this detail to the text (see lines 523, 543, 569). The genomes have both been polished iteratively using Pilon with short reads five times.

Line 449: There is an error statement in the text here.

Many thanks. We have now addressed this.

Line 542: The link to the Github repository is broken.

Line 593: The link to the Github repository is broken here as well.

We apologise for the inconvenience. We have fixed the accessibility permission to the GitHub repositories.

Reviewer #2

The paper by Talenti and colleagues presents a bovine graph genome that integrates the current Hereford-based reference sequence and structural and single nucleotide variants detected from different breeds of taurine and indicine cattle. It is increasingly recognized that the use of a linear reference genome does not appropriately capture genetic diversity within a species. Talenti and colleagues augment the taurine reference genome with variants detected in indicine cattle to make the reference sequence more informative for cattle from "underrepresented" breeds. They detect a total of 202 Mb non-reference sequences.

The paper presents four different vignettes. The assembly of two African cattle genomes is followed by the construction of a multi-assembly graph, subsequently, variant calling from an augmented reference structure is presented, followed by an analysis of ATACseq peak calling from an augmented reference.

In the first vignette, the authors assemble the genomes of two African cattle using PacBio CLR reads. Although the availability of genome assemblies for African cattle is a valuable resource, the lack of description of these assemblies tampers enthusiasm. Scaffolding was done using either the

"previously published genomes" (which ones???) or Bionano data. The quality of the assemblies is reasonable as indicated by N50 values and BUSCO scores. However, the assemblies are less contiguous than those recently generated by trio binning or the use of HiFi data. The supporting files describing the assembly process are merely a sequence of shell scripts and should be revised to present the process in a readable way (or at least a synopsis should be presented).

Thanks. There are inherent challenges to working with cattle breeds from low and middle income countries, not only in terms of funding as raised by reviewer 3 but also in terms of collecting Nagoya protocol compliant, high-molecular weight samples in the field that can be transferred to sequencing centres that are unfortunately almost exclusively not in Africa. But as the first assemblies of African cattle, with good N50s and whose quality metrics are largely in line with those of other assemblies, we firmly believe these genomes are invaluable resources to help fill in the current gaps in global breed diversity. We have now substantially edited Supplementary Documents 1 and 2 to make them easier to read and comprehend.

More details about the assembly need to be presented in the results section, e.g., the sex of the animals, the number of contigs and scaffolds, as well as the assembly length and quality.

Thanks. We have now expanded the results section to address these points (see lines 124-135, 143-147).

Moreover, according to the discussion, the assemblies were created from cattle of African taurine and taurine x indicine ancestries. This should be presented in the results section.

We apologise for the confusion. The Ankole is also referred to as a key representative of "Sanga" cattle that are indigenous African indicine cattle but with potential ancestral introgression from taurine breeds. We added the lineages of the two individuals (Taurine and Sanga) in the results section at lines 124 and 125. We have also changed the discussion to "an ancient stabilized cross between indicine and taurine breeds" (line 408).

The ancestries of the two assemblies need to be related to the findings presented in Figure 2.

We have now performed an extensive rewriting of the results and changed several figures, including Figure 2, trying to connect better to the results obtained.

In the second vignette, the CACTUS aligner is applied to detect sequences missing in the current bovine reference genome. The paper reports 202 Mb non-reference sequences from two novel African cattle assemblies and two previously generated genome assemblies from a Angus x Brahman cross (<https://doi.org/10.1038/s41467-020-15848-y>). The way the results are presented is confusing, particularly Figure 2, and needs a thorough revision. Are the units correct (1e8)? My interpretation of Figure 2 is that the five horizontal bars represent the sizes of the assemblies and the vertical bars represent the amount of sequences that are shared between the assemblies. What is the sum of all intersections presented in this Figure? Are the values supposed to sum up to 202 Mb or to 2995 Mb (Supplementary Table 3; why have some numbers in Table S3 digits?), or to a totally different value. From my understanding, Figure 2 indicates a problem with the Angus assembly. According to data curated at NCBI (GCA_003369695.2), the Angus assembly is only slightly smaller than the Brahman assembly (2580 vs 2680 Mb), and the difference is even lower, when considering only autosomal sequences (2468 vs 2478 Mb). A high vertical bar indicates a large amount of sequences is present in all but the Angus assembly. This is weird, because it is clearly too small to be attributable to the X chromosome (~140 Mb). Also, the horizontal bar indicates the size of the Angus assembly is less than half of Hereford, which seems not reasonable. As one of the African cattle genomes originates from a

taurine x indicine cross, I'm wondering if this assembly contributes more non-reference sequences than the other.

We performed extensive changes to the method used to detect the novel portions of the genomes that we hope might help in making the section easier to understand and follow through. The analyses are now referring to the non-reference portions of the genomes (i.e. the non-ARS-UCD1.2 sequences) that are not unusually repetitive. The initial amount of novel bases has increased to >250Mb but this includes single nucleotide variants and small indels. We now go through several refinements that includes excluding portions that were potential misassemblies, that were in regions of potentially lower quality (close to telomeres/gaps) and that are unusually repetitive. The change in the approach also removed the ambiguity leading to the floating-point estimates: the previous version calculated the novel sequence using MAF alignments referenced to each genome separately, leaving margins for change due to repetitive elements. This version, referenced specifically to the Hereford assembly, removed the ambiguity leaving a single non-reference value. After these adjustments the amounts of novel sequence is now more reflective of the expected tree, with the indicine Brahman and Ankole having the highest amount of novel sequence relative to the Hereford, followed by the N'Dama and finally the Angus (see sections "Detection of non-Hereford sequence" and "Gene content in the novel sequences" at line 151-204).

In the third vignette, the authors augment the Hereford-based reference with non-reference sequences to create a graph genome. Compared to a recently published analysis of a bovine genome graph (<https://doi.org/10.1186/s13059-020-02105-0>), the results shown here are presented superficially and don't provide novel insight. It is unclear how the graph was created (methods: "we aligned the genomes chromosome by chromosome..."). Is this approach different to the CACTUS-approach applied earlier? Did the graph contain the 202 Mb non-reference sequences that were detected previously, or only a subset of these? Did the graph contain only autosomal sequences, or also unplaced contigs, and sex chromosomes?

We thank the reviewer for pointing out that the sentence wasn't clear. We rephrased the different section (section "Constructing the graph" at line 205) to make it clearer, specifying how the graph used for the resequencing data analyses is only autosomal and, therefore, containing a subset of the novel sequence previously detected. With respect to the previous graph genome the reviewer refers to, we believe there are some important differences. For example, the previous graph was just an augmentation of the Hereford genome using variants called from short-read sequencing data, i.e. not using novel assemblies. It was also restricted to both the incorporation and downstream study of European taurine breeds, whereas we were interested in studying how more representative graph genomes may help reduce the biases inherent in using a Hereford reference when studying more divergent African and Bos Indicus cattle. Also, as the authors state in the manuscript they largely ignored SVs due to the lack of validation data, whereas we provide substantial analyses on the concordance of vg SV calls using independent optical mapping, and now Delly, calls. We have further expanded these sections (see lines 323 to 365).

The way VG5p was created from the 11 M short variants remains elusive. Previous studies showed that variants need to be prioritized before adding them to a graph (<https://doi.org/10.1186/s13059-018-1595-x>). Also, a haplotype index may be considered to add only biologically plausible paths in the graph. The comparison between VG and BWA is not really appropriate, because VG may provide better mapping accuracy from an empty graph than BWA. Thus, findings need to be related to VG1. The paper shows that VG5p did not really perform better than VG5 (Figure 4), possibly because the variants were not prioritized accordingly. Would VG5 enable better mapping accuracy than a graph constructed from only 11M variants (no structural variants added) that were appropriately prioritized?

We have now added a graph constructed from just the Hereford genome augmented with the 11M variants as the reviewer suggests (VG1p). The results show that although the number of perfectly mapped reads is higher than the Hereford only graph, it is less than the VG5 and VG5p versions. Also, the total number of mapped reads (perfect or not) is not actually altered with respect to the VG1 genome, suggesting that the increase in mapped reads is fundamentally related to the presence of novel sequence in the VG5 and VG5p graphs. See lines 205 to 272.

The fourth vignette deals with ATACseq peak calling. It seems that this analysis is largely disconnected from the previous sections, because it relies on a linear reference sequence that has been expanded with non-reference sequences. A similar approach was presented recently for ChIPseq calls (<https://doi.org/10.1186/s13059-020-02038-8>)

Previously, we had tried to use the development branch of graph_peak_caller (https://github.com/uio-bmi/graph_peak_caller), to analyse the ATAC-seq data using the VG5p graph. However, the software seemed to not cope well with the graphs generated through cactus. For this reason we focused on any merits to expanding the current reference to capture missing sequence, an approach that will be compatible with all current analysis tools.

In this version, we have worked on better integrating this analysis into the paper as a whole. In particular, we refined the extraction of the contigs, using a much more stringent selection of the candidate sequences. This reduced the total sequence considered to ~20Mb of contigs that excluded highly-repetitive sequences. This reduced the impact of lower quality portions of the component genomes, improving the accuracy of the peak calling. Moreover, we performed a gene prediction on these sequences and show that these novel genes are often enriched for ATAC-seq peaks at their TSS. Consequently, including such novel sequence has the potential to provide novel biological insights that would otherwise be missed. See lines 373-405.

The discussion somewhat fails to put the work into context as it neglects a broad body of literature on recently created whol-genome graphs.

We tried to further expand the introduction and state of art on graph genomics (see lines 93-97).

Reviewer #3

The authors have created pan-genomes based on reference assemblies from 5 different cattle breeds, and showed the benefits of such a graph genome. They assessed variant calling including larger structural variation, and ATAC-seq benefits. Showing the benefits of graph genomes for functional analysis. It is a very extensive and well conducted study in a rather new field of interest to the community.

General comments

I believe the introduction should not include the breeding part (L77-81), but keep focus on the actual benefits of a graph genome. For breeding a graph genome is not directly needed (although it may benefit from findings). I guess the main benefit of graph genomes are the detection of structural variants and genetic differences that are breed specific, and hence could aid detection of causal variants for e.g. disease resistance. So the advantage is really on the functional detection directly, and on breeding indirectly after discovery of functionally important variants, but only if a breeding

program including genomic selection is in place, which would be the major challenge in African countries.

Thanks for the suggestion. We have now edited the introduction accordingly (see lines 101-103).

Is a pan-genome better than a breed (or line) specific reference genome? In order to create a pan-genome, you need to make a de novo assembly of other breeds, in the future many breeds will have their own reference genome to align against, will a pan-genome be needed then? I believe that SV calling is also improved when using a breed specific ref genome. The comparison of for instance N'Dama reference vs VG5 has not been made (or I missed it), and would be valuable to add. I think the pro's and con's of a breed specific reference genome versus a graph genome should at least be discussed.

The issue of any single linear reference genome, whether specific to the breed being studied or not, is that it does not include the variation observed across animals. As we highlight in this study, perhaps the greatest advantage of graph genomes is the ability to type structural variants more accurately. For example, if an animal being studied has an insertion relative to the reference being used, this is difficult to accurately infer from standard approaches, even for breed-specific genomes, given the sequence is missing from the reference and cannot therefore be mapped to. For this reason, when using Delly v2 in this study we could not properly call insertions. In contrast, as a graph genome can incorporate the known diversity at this position it can be explicitly genotyped. There may sometimes be merits to building graph genomes for specific breeds, however, like for genotype imputation reference panels, incorporating as much of the known species diversity into the graph will likely be most effective. This is because SVs that are rare in the breed of interest may be common, and therefore captured, in other breeds. Using a diverse set of animals to construct the graph will simply ensure the greatest amount of diversity is incorporated and can be genotyped. So even if a breed-specific genome exists, supplementing it with the diversity observed across animals in the form of a graph genome is expected to improve SV calling.

We have now highlighted this in the discussion at lines 438 to 441.

For the variant calling freebayes and GATK were applied to ARS1.2 genome, hence they can be directly compared to VG1. Looking at the results in Supplementary document 3 VG1 performs poorer than the commonly used GATK (but better than freebayes). Please also comment on that. Any hints on why this happens? More importantly what does it mean for using graph genomes in general?

Yes, currently the primary benefits of graph genomes is around the calling of SVs, and Freebayes and GATK are likely more effective at calling shorter variants due to their more advanced approaches adopted to call these variants. We have now more explicitly commented on this at lines 468-471 *“Moreover, variant calling currently relies on a pile-up approach, which is arguably less sophisticated than methods implemented by GATK or FreeBayes, that likely helps explain the good performance of traditional tools at calling SNPs and small indels”.*

You focus on the allelic balance but there are other features, e.g. transition-transversion drop for unknown angus variants for VG5, variant quality for VG1 low

Yes we had deliberately focused on allelic balance as we believe it is one of the higher level metrics that impacts variant calling and that is most easily compared between approaches. Variant quality is for example more difficult to directly compare between GATK and vg, but we provide these metrics for info in Supplementary document 3. Given the consistencies observed for the other breeds we believe the difference observed between the transition/transversion ratios for the Angus samples is likely a combination of the sample quality and the simpler SNP caller implemented by vg

(see above). GATK and freebayes simply appear to perform better at SNP calling, though as vg is under active and intense development, newer versions might address this. We have now added a comment to the results to describe this (lines 267-271).

The results section can benefit from some sub-sub headings to aid the reader when switching topic within a sub-section.

Thanks for the suggestion. We have now added several sub-sub-sections within the paper to make it easier to follow.

The links to the github are not working because the folder is not there yet, please make sure it is available before publishing.

We apologise for the inconvenience, we have now fixed the access permissions to the repository.

Line by line comments

L61 I think the wording should be changed a bit, as the 1000 bull genomes project actually also facilitates indicus breeds (see presentation on their website), does variant calling for indicus specific variants as well as a combined indicus and taurus set. Yes there are far more European breeds in their resource, but that has to do with more funding being available for those breeds.

We thank the reviewer, we have adjusted this text accordingly (lines 75-76).

L80-81 Do you really believe a reference population augmented with local breeds is the main factor to drive selection in Africa? There is an indicus HD SNP array that will capture genomic relationships to circumvent the lack of accurate pedigree recording. Accurate trait recording would be essential to start a good breeding program. To organise the structure and centralise it, that is the main difficulty for breeding.

We agree that there are many challenges to improving livestock production in Africa and have now removed this section.

I guess an pan-genome including African breeds would be useful to identify causal variants for e.g. disease resistance as mentioned later on. Please see my general comments to improve the introduction in this respect.

Addressed above. Thanks.

L199 Change 'with 163M and 194M nodes and edges' into 'with 163M nodes and 194M edges'

Thank you, this is now changed (line 218-219).

L240-242 Freebayes is not designed to call large variants (not larger than the read length). For GATK haplotypcaller also only SNP and indels can be called. So not fair to compare. However, it is worthwhile to mention that VG can call such large variants together with short variants. Just wondering how accurate they are. To properly address vg accuracy for calling large variants you will need to compare the results to some SV callers that can call variants of similar size and type.

As suggested, we have now also run Delly v2 on the different samples. There are some difficulties directly comparing the different approaches: one directly genotypes a known sequence, whereas

the second infers the presence from misassembled reads. This provides a contrast of implicit and explicit SVs which might affect the comparisons between the signals. Nevertheless, we can see that, overall, VG appears to call SVs that are more consistent with the independent optical mapping data, suggesting the graph approach appears to improve the reliability of SV calling. See lines 323-365.

L252 Change NDama into N'Dama

Thanks. We have now corrected this (line 285).

L274-275 Wasn't it the VG5p graph that was used for the SV calling (with vg), rather than the Hereford (VG1), as indicated in line 246

Yes, the variant calling has been performed using the VG5p graph. Nevertheless, the coordinates system used by the variant caller relies on the positions on the Hereford genome. We have now rephrased the sentence using "Hereford genome used as backbone for the graph" to make it less ambiguous (see lines 306-307).

L352-354 Is it just insertions and deletions that can be called or also more complex SV like inversions?

vg is capable of calling insertions, deletions and some more complicated types of SVs such as inversions and very large complex SVs (see <https://genomebiology.biomedcentral.com/articles/10.1186/s13059-020-1941-7#Sec12>). See lines 765 to 781 for more details.

L416 Please define OM

Thank you, it is now introduced at line 280 and used consistently across the manuscript.

L449-450 Please fix reference error: Error! Reference source not found
L506+542 The github links give errors, going to github.com/evotools shows that the folder for this paper is not there

We apologise for the inconvenience, we have now fixed these.

L558-560 Sentence 'Samples were then sequenced on a Illumina HiSeq X Ten at the Edinburgh Genomics sequencing facility.' Repeated, please remove one.

Thanks for spotting. Repetition now removed (line 695).

L617 With this one heterozygote allowed it is not clear to me what you mean. Does at least one individual have to be heterozygote? What if the only individual carrying the variant is homozygous (if that even occurred). I guess you want to be certain that you have at least one carrier with sufficient evidence hence the minimum allele count of 5. I guess if my reasoning is correct you can simply add 'at least one heterozygote with sufficient evidence for the alternative allele'.

Thank you, we implemented the suggested change (line 752-753).

Figure 2A It took me a while to understand the figure. Perhaps a table would be easier to understand

Thank you, we changed the figure from an upset plot to a SuperExactTest plot, which we believe is easier to read, and provided an improved description in the legend (now Figure 3A).

Figure 5A What do the greyish rectangles indicate in between?

We have now edited the caption of the figure to better explain the meaning of the different colours of the boxes.

Supplementary material

- Please add table & figure descriptions to explain the viewer what is in there.

Thank you, we have now added the captions of the different figures, tables and documents to make them more accessible (lines 1073-1125).

- Figures do not always fit on a page, please adjust.

Thank you, we have adjusted the sizes of the figures.

- In table 6 under nSeq is that the number of contigs? If so please change into the more informative 'nContig'

Thank you, we have now changed the header of the table (now Sup. Table 5).

- In table 10 the final bargraph does not include a bar for ARS+ which is indicated in the legend

Many thanks, we have now added the missing bars (now Sup. Table 11).

REVIEWER COMMENTS

Reviewer #1 (Remarks to the Author):

Summary: In this revision, Talenti et al. have addressed many of my major and minor comments. It is quite clear that the authors invested substantial time in revising the manuscript and I feel that the results are more clearly presented and discussed in this version. I still have major comments for the authors to address, but I believe that the quality of the manuscript has greatly improved.

My major comments (in no particular order):

Figure 3B: What is the significance of the dendrograms? Were these derived from the whole genome alignments? Do the dendrograms change between top and bottom portions of the figure? Also, please provide a distance legend to present the significance of the separation of each genome into clades.

Supplementary Note 1: There are still error statements littered throughout submitted materials. I suspect that there may be some issues with the reference manager software used by the authors. I recommend that these be corrected in subsequent revisions.

Line 280: I am still quite skeptical of the quality of the N'Dama assembly. My skepticism arises from the unusual reliance on reference-guided assembly and the continued prevalence of FRC errors in the assembly (seen in the last FRC plot of Supplementary Note 1). I truly believe that using the optical mapping data of the N'Dama individual that was the offspring of the reference animal would have provided less bias in the assembly of the autosomal sequence than the approach taken by the authors, but I understand that a major "redo" of the assembly would take months of work. I suspect that a suitable compromise would be if the authors provided Bionano OM "conflict" estimates from alignments against their original contigs. This would provide a reasonable assessment of the structural quality of their initial set of contigs prior to reference-guided assembly. If regions that do not have conflicts with the OM data are being corrected by Ragout2 (now "Ragtag" I believe), then that means that true structural variants in N'Dama are being mistakenly corrected. If these instances are insignificant, I would be more convinced of the suitability of the assembly for subsequent SV analysis.

Line 428: In light of the relatively poor performance of the VG1p graph, I would like to see a bit of discussion here regarding the sole use of SNP and INDEL variant incorporation into genome graph references. This is still a major source of debate within the community, and it appears that the authors' results suggest that known smaller polymorphic sites have been sufficiently handled by read alignment (and graph alignment) software and provide little value when added to genome graph references. This is a major point in favor of this study that should be highlighted in this section.

Reviewer #2 (Remarks to the Author):

The authors revised their manuscript, taking constructively into account comments from three reviewers. This led to some major changes in the manuscript including refined methods to detect and characterize non-reference sequences. The authors now report that their pangenome contains 116 Mb of additional sequence compared to the Hereford-based reference genome. These sequences were detected in 62,337 nodes longer than 60 bases and included 20.5 Mb non-repetitive sequences that were novel when compared to ARS-UCD1.2. The sharing of the non-reference sequences among the studied assemblies agrees well with ancestry differences among the assemblies. The gene content of the non-reference sequences was predicted using three bioinformatic approaches.

Talenti et al. now employ revised and refined approaches to quantify and characterize non-reference sequences and assess the gene-content of the non-reference sequences. These approaches as well as

the results presented in their revised manuscript mirror those from a recently published bovine pangenome graph (<https://doi.org/10.1073/pnas.2101056118>). The similarity of results between both studies is expected as three of the five assemblies considered in Talenti et al. were used by both studies. In spite of similar aims, methods and results, there is no discussion about similarities and differences between the two bovine pangenome graphs. Moreover, the reference to Crysanto et al (line 97) is misrepresentative, as their graph was constructed using non-reference sequences >100 bp rather than large differences between assemblies. Talenti et al. also considered non-reference sequences of a similar size from the assemblies (line 164). The 11M variants added to the graph in the present manuscript were smaller variants detected from short reads, not structural variants which are more important in a pangenome.

It's unclear how VG5 was constructed. As the section about graph construction is presented immediately after a detailed description of the gene-content of the high-quality non-reference sequences, I assume that the 16,665 sequences with length >60 bases containing a total of 20.5Mb non-reference sequences were used to construct VG5. Is this correct?

As mentioned in the previous review, the construction of variation-aware genome graphs (line 206 ff) seems to neglect recommendations on variant prioritization (<https://doi.org/10.1186/s13059-018-1595-x>). The ancestry of the 294 samples is unclear. Do they indeed represent "global breed diversity"? Referring to Dutta et al. is not enough, as the ancestry of the samples is critical for variant prioritization to enable constructing informative graphs. Are the 11 million variants detected in the 294 cattle also representative for the samples from Angus, NDama, and Sahiwal that were used for mapping and variant calling evaluations? Previous studies (in different species including cattle, <https://doi.org/10.1186/s13059-020-02105-0>) showed that adding random variants to a genome graph does not lead to any improvements over linear mapping. Talenti et al. report that VG1p does not lead to mapping improvements over VG1, and this is likely because the variants were not prioritised. In addition, a haplotype index may further improve mapping and resolve mapping ambiguity (<https://doi.org/10.1093/bioinformatics/btz575>). Talenti et al. state in the discussion that graph genomes do not lead to significant improvements in small variant calling (l. 428). This is a contradiction to what is presented in their results particularly to Figure 4b,c. Improvements are likely even higher when variants are prioritized appropriately.

Did ARS-UCD1.2+ also increase mapping ambiguity?

Authors indicate in their revised version that Angus samples from public databases present a lower Ti/Tv ratio (line 271). What does this mean/imply? Is there a problem with these samples? Or is the Ti/Tv ratio low, because the graph does not permit accurate variant detection in taurine samples? Can this be tested with other samples from taurine ancestry?

Is the "54%" in line 228 (apologies for missing this in my first review) correct? An improvement of 54% in perfectly mapping reads seems exceedingly high.

Reviewer #3 (Remarks to the Author):

The manuscript has been extensively revised, and my comments have been addressed.

I am satisfied with the responses to my comments on structural variation calling/line specific reference genomes and pleased to see a comparison with Delly is included. However, the arguments given in the rebuttal do not come across properly in the manuscript.

From the rebuttal I understand that VG can only detect SV present in the graph and thus no 'novel' SV from an independent re-sequenced individual, correct? This would explain the large difference in detected SV by both methods. Delly is known to be outputting a large amount of SV with varying confidence to let the end user decide on filtering criteria, so yes many should be filtered and cannot be

confirmed by other methods. In the manuscript the focus is given to the specificity of VG which is then not really fair. It is good to highlight that SV you find with VG are more likely to be true SVs as most are confirmed by Delly and OM (which makes sense as they have already been observed in individuals used to construct the graph), however, that it cannot exploit split read and discordant reads to detect novel SV.

So I agree with the argumentation in the rebuttal where this is explicitly indicated, but I like to see those arguments better reflected in the manuscript. Rather than claiming VG is a good SV caller due to its higher specificity compared to Delly (implicitly suggesting Delly is a poor SV caller) it can be nuanced.

Below are a few sections in the manuscript that need reformulation with respect to the above mentioned comment.

Line 350-355 & Line 433-435 You are focussing a lot on the specificity here which makes VG the better SV caller, however, you ignore the fact that Delly detected a much higher number of deletions confirmed by OM (3,186 vs 1,887). Delly is known to be very sensitive and outputting as many SV as possible to leave the filtering to the user for it specific purpose.

Line 356-360 This doesn't feel like a fair example when indicating that Delly didn't detect it. You use an insertion as example here. Insertions were filtered for Delly because they had imprecise breakpoints (line 331-333).

Figure 4D: I don't understand the numbers given at the intersection, shouldn't they be equal for the methods included in the intersection?

Textual comment

L603 change 'We the applied...' to 'We then applied...'

L652 I believe '<60bp' should be >60bp

RESPONSE TO REVIEWERS' COMMENTS:

Reviewer #1 (Remarks to the Author):

Summary: In this revision, Talenti et al. have addressed many of my major and minor comments. It is quite clear that the authors invested substantial time in revising the manuscript and I feel that the results are more clearly presented and discussed in this version. I still have major comments for the authors to address, but I believe that the quality of the manuscript has greatly improved.

My major comments (in no particular order):

Figure 3B: What is the significance of the dendrograms? Were these derived from the whole genome alignments? Do the dendrograms change between top and bottom portions of the figure? Also, please provide a distance legend to present the significance of the separation of each genome into clades.

The tree represented in the previous images simply reflected the known relationship between the breeds and was used to indicate the default positioning of each breed in the picture. This tree was used for display purposes only, since the generation of the alignments are computed pairwise by AliTV using lastz, ignoring the phylogenetic tree. To avoid confusion, we have now removed the tree from the figure.

Supplementary Note 1: There are still error statements littered throughout submitted materials. I suspect that there may be some issues with the reference manager software used by the authors. I recommend that these be corrected in subsequent revisions.

We apologise for the inconvenience. We are unsure what might be causing the problem, since these errors didn't appear for us. To hopefully avoid further issues, we changed the reference manager from Mendeley to Zotero.

Line 280: I am still quite skeptical of the quality of the N'Dama assembly. My skepticism arises from the unusual reliance on reference-guided assembly and the continued prevalence of FRC errors in the assembly (seen in the last FRC plot of Supplementary Note 1). I truly believe that using the optical mapping data of the N'Dama individual that was the offspring of the reference animal would have provided less bias in the assembly of the autosomal sequence than the approach taken by the authors, but I understand that a major "redo" of the assembly would take months of work. I suspect that a suitable compromise would be if the authors provided Bionano OM "conflict" estimates from alignments against their original contigs. This would provide a reasonable assessment of the structural quality of their initial set of contigs prior to reference-guided assembly. If regions that do not have conflicts with the OM data are being corrected by Ragout2 (now "Ragtag" I believe), then that means that true structural variants in N'Dama are being mistakenly corrected. If these instances are insignificant, I would be more convinced of the suitability of the assembly for subsequent SV analysis.

Sorry, we believe the reviewer may be confusing tools. The tool Ragtag (<https://github.com/malonge/RagTag>) to which the reviewer refers, actually used to be called RaGOO. Whereas the tool used in our analysis was Ragout (<https://github.com/fenderglass/Ragout>). We admit it is confusing. As the reviewer suggests we had considered whether to scaffold the genome with the optical mapping data from its son. However, the bionano scaffolding software explicitly assumes the optical mapping data is from the same individual, and therefore has the potential to also introduce errors where there are inconsistencies between the two. In contrast Ragout does not make the same assumption, as it implicitly assumes the assemblies are from different animals. Ragout has been demonstrated to generate chromosome-scale assemblies as accurate as those using Bionano and HiC data (see

Kolmogorov et al. *Genome Res.* 2018 28 1720-1732) and has been shown to be able to preserve structural variants present in the assembled contigs.

As suggested we have now scaffolded the NDama with the optical mapping data from its son and compared the two assemblies. As can be seen in the dotplot below, the genomes are highly collinear, though with three chromosomes split across scaffolds in the bionano assembly which are successfully scaffolded at the chromosome level when using Ragout (chromosomes 8, 9 and 23).

Figure 1: Dotplot comparison of the ragout (x axis) and bionano (y axis) assemblies. Those chromosomes successfully scaffolded by ragout but not the bionano data are indicated by purple boxes

The two approaches did perform a different number of cuts, with ragout cutting more contigs than with the optical mapping data, but this is in part due to the different resolutions of the two methods. Whereas Ragout relies on alignments and can potentially detect very small misassemblies, optical mapping is limited by the labelling resolution of several Kb. Looking more specifically at the changes made by the bionano data, we can see that 87 out of 122 cuts introduced by the optical mapping had a cut on the same contig introduced by ragout. Of these, 48 were less than 10Kb apart, 75 were less than 50Kb apart and 78 were less than 100Kb apart, potentially referring to the same misjoin within the contigs detected by both methods (as discussed there is expected to be some fuzziness in the breakpoint locations). Below, is an example of a conflict detected by the optical mapping data that was also corrected by ragout, with a difference in the breakpoint of 2Kb.

Figure 2: Bionano data at an example location corrected by both approaches

Consequently 35 cuts were made by the Bionano data in regions where no change was made by Ragout in close proximity. These involved 23 contigs that in total spanned a total of 36.18Mb, 19 of which were included in the chromosomes scaffolded by ragout. These overlapped just 75 regions out of 3,697 identified as specific to the N'Dama genome and were only 56,816 bp long in total, out of 3,684,492 bp of N'Dama-specific sequence detected. Most of the cuts introduced by the optical mapping on these regions are at the terminal part of the contig and as can be seen in the dotplot the bionano data is sometimes trimming off the repetitive telomeric regions of chromosomes. If we just sum the amount of sequence at these trimmed regions if they make up less than a fifth of a contig (but sum the whole contig otherwise) they correspond to just 2.46Mb of sequence. Of these, 29 (1.92 Mb) have been placed in a primary scaffold, and overlap 26,553 bp across 26 N'Dama specific regions (further reduced to 17 regions and 4,494 bp when accounting for redundant regions across multiple genomes). Consequently, if we assume all the changes the Bionano data is making are correct, and these aren't for example conflicts due to the maternal haplotype, then it is only a small amount of the genome and involves very little NDama specific sequence.

Given the large number of assemblers available and possible scaffolding approaches we admit there are many ways that could have been adopted to assemble these genomes, each with their own advantages and disadvantages. But the data suggests the approach we adopted has not led to a high level of assembly errors. All underlying data has also been made available for others to use. In response to the reviewer's point we changed the methods (lines 564-569) as follow:

" Although an alternative strategy to scaffolding this genome would have been to use Bionano data from its offspring we did not find using this approach substantially altered the genome or the conclusions of this study. Unlike Ragout, the Bionano scaffolding did not successfully generate chromosome-level scaffolds in all cases, and we estimated that using the optical mapping approach would lead to less than 30Kb of N'Dama-specific sequences being altered among the primary scaffolds."

Line 428: In light of the relatively poor performance of the VG1p graph, I would like to see a bit of discussion here regarding the sole use of SNP and INDEL variant incorporation into genome graph references. This is still a major source of debate within the community, and it appears that the authors'

results suggest that known smaller polymorphic sites have been sufficiently handled by read alignment (and graph alignment) software and provide little value when added to genome graph references. This is a major point in favor of this study that should be highlighted in this section.

We thank the reviewer for the suggestion. We expanded the discussion in sections in lines 442-454 to address this point as suggested.

Reviewer #2 (Remarks to the Author):

The authors revised their manuscript, taking constructively into account comments from three reviewers. This led to some major changes in the manuscript including refined methods to detect and characterize non-reference sequences. The authors now report that their pangenome contains 116 Mb of additional sequence compared to the Hereford-based reference genome. These sequences were detected in 62,337 nodes longer than 60 bases and included 20.5 Mb non-repetitive sequences that were novel when compared to ARS-UCD1.2. The sharing of the non-reference sequences among the studied assemblies agrees well with ancestry differences among the assemblies. The gene content of the non-reference sequences was predicted using three bioinformatic approaches.

Talenti et al. now employ revised and refined approaches to quantify and characterize non-reference sequences and assess the gene-content of the non-reference sequences. These approaches as well as the results presented in their revised manuscript mirror those from a recently published bovine pangenome graph <https://doi.org/10.1073/pnas.2101056118>. The similarity of results between both studies is expected as three of the five assemblies considered in Talenti et al. were used by both studies. In spite of similar aims, methods and results, there is no discussion about similarities and differences between the two bovine pangenome graphs.

As the reviewer points out Crysanto et al. have also published a nice manuscript building a cattle genome graph. But there are many important differences between the studies:

- 1. An over-riding aim of our study was to generate a more representative reference genome resource that included data from low and middle income country (LMIC) cattle breeds, given the disproportionate importance of livestock to LMIC economies. We therefore generated the first two African cattle reference genomes of both taurine and indicine ancestries, and incorporated them into a graph genome with shorter variants called across 294 samples including 82 from Africa. Crysanto et al. had no novel African or LMIC samples, from either long read assemblies or short read data, in their graph which, as is the case for most cattle studies, is focused on European breeds. The novel assembly in their paper was from the Original Braunvieh European breed (it did include the public US Brahman and the yak genome).***
- 2. Beyond the diversity of samples included, even the construction of the genome graphs was very different between the two studies. For example, in our study we used the reference-free aligner cactus and vg whereas Crysanto et al. used the reference based minigraph. Minigraph is a very nice tool, with a key advantage being it is fast, but it does have some important relative limitations, most notably its reference bias (which in the case of Crysanto et al. was the Hereford genome). We were deliberately trying to build a genome that is not biased towards the European reference. Minigraph also has limitations with respect to missing smaller variation and being lossy. For more details see <https://ekg.github.io/2019/07/09/Untangling-graphical-pangenomics>. Consequently whereas we had augmented our graph with short variants called across 294 further animals,***

and compared the performance, Crysanto et al. had only incorporated larger differences between the six assemblies into their minigraph graph.

- 3. Crysanto et al.'s analysis of structural variants (SVs) was restricted to identifying sequence differences between the six assemblies (relative to the Hereford) and aligning long HiFi reads from a single crossbred animal (indicine Nellore X taurine Brown Swiss) to the graph to identify where these between assembly differences were supported by the reads from this single animal. In contrast we explicitly investigated how well the diverse vg graph could call SVs in independent samples from divergent indicine and taurine breeds, and importantly used data from a completely different approach, optical mapping, to identify where the SVs called were consistent. Following the reviewers' suggestions we have also included a comparison to SVs called using completely different software Delly. So the study of SVs is very different.*
- 4. We not only looked at the DNA-seq mapping rates across a more divergent set of breeds than Crysanto et al. but also for example looked at the mapping rates of ATAC-seq data also from divergent breeds and highlighted how the Hereford is likely missing regions of open chromatin.*

*These are just some illustrative examples of how the analyses differ, with for example a range of further analyses in our manuscript, such as comparisons of different graph construction approaches. These shouldn't be read as criticisms of Crysanto et al., they were simply doing different analyses with different aims. For example, in their paper is a study of differential expression in *M. bovis* infection that is not done in our study. So we believe it would be misleading to suggest the aims, methods and results of the two studies are largely the same. Unlike Crysanto et al. the aims of our study was to produce a genome graph better representing global breed diversity, to compare the performance of different graphs, to assess the calling of SVs using optical mapping approaches and Delly and to look at other omics data. We did this constructing the genome graph in a completely different way to that of Crysanto et al.. The reviewer is correct that the approaches used for one analysis added at the revision stage, the detection of novel gene sequences, is similar between the two studies and in our revisions we had cited Crysanto et al. in this section accordingly. But we should point out that Crysanto et al. submitted their manuscript to PNAS over 8 weeks after this manuscript was submitted to Nature Communications, so it simply wouldn't have been possible for us to have made comparisons between the two studies at the time. Following the reviewer's suggestion we have now added a short description of some differences between the studies in the discussion at lines 422 to 424.*

Moreover, the reference to Crysanto et al (line 97) is misrepresentative, as their graph was constructed using non-reference sequences >100 bp rather than large differences between assemblies. Talenti et al. also considered non-reference sequences of a similar size from the assemblies (line 164).

As the reviewer says, Crysanto et al. had only incorporated larger non-reference (>100bp) differences in their graph. In our VG5 and VG5p graphs we include the entire range of changes from millions of SNPs, including those between the assemblies, to small and large structural differences between the genomes. The 60bp filter that we believe the reviewer is referring to was only applied when calculating the amount of high-quality novel regions in the different assemblies. The shorter changes (<60bp) are still in our main graphs (as specified in lines 162-163). This type of resolution is not achievable when using minigraph (as used by Crysanto et al.), as stated by the developers themselves with the limitations section of the GitHub page stating "the software falls short in detecting, for instance, graph consisting of many short segments (e.g. one generated from rare SNPs in large populations), minigraph will fail to map query sequences". The cactus approach, while more computationally intensive, can detect these types of variants, is not reference based, and in fact

allowed us not only to detect novel large regions but also highly divergent haplotypes that might be equally interesting.

Considering the variants included in the short-reads sequencing aligned to the graph genome, we considered all the variants (including SNPs and small InDels) derived from single-chromosome alignments (i.e. all chromosomes 1 from the five assemblies together), and even augmented our graph with 11 million additional variants called across 294 animals from divergent breeds. We have now explicitly stated this on line 213.

Finally, we amended lines 95-98 specified by the reviewer as follow:

"In livestock, the use of graph genomes has so far been restricted to studies simply incorporating variants from short read sequencing data into the Hereford reference or to only large (>100bp) differences between non-Hereford assemblies with the ARS-UCD1.2 reference genome."

The 11M variants added to the graph in the present manuscript were smaller variants detected from short reads, not structural variants which are more important in a pangenome.

We agree that the shorter variants are mostly SNPs and not SVs and the SVs are more important to the results. We also stated this in the manuscript, e.g. see from line 440 "In this study we illustrate that the use of the graph cattle genome does not lead to substantial improvements in the calling of SNPs and small indels, even when large numbers of them are integrated into the graph". But as raised by reviewer 1 "I would like to see a bit of discussion here regarding the sole use of SNP and INDEL variant incorporation into genome graph references..." this is an important point which otherwise couldn't be addressed without making these direct comparisons to graphs incorporating different sets of variants.

It's unclear how VG5 was constructed. As the section about graph construction is presented immediately after a detailed description of the gene-content of the high-quality non-reference sequences, I assume that the 16,665 sequences with length >60 bases containing a total of 20.5Mb non-reference sequences were used to construct VG5. Is this correct?

We thank the reviewer for pointing out that we did not add a sentence to explain the origin of VG5 in the material and methods. VG5 was constructed performing chromosome-by-chromosome alignments using CACTUS (i.e. all CHR1 together, all CHR2 together and so on). These alignments have then been converted to vg format using hal2vg, and indexed all together using the VG indexing workflow (XG generation, pruning and then GCSA generation). So there was no filtering to just regions greater than 60bp in length, rather VG5 includes all differences between the genomes as stated in the response above. We added the following sentence at lines 696-697:

"We first aligned the five cattle assemblies using CACTUS chromosome-by-chromosome (i.e. all chromosomes 1 from the five genome together)."

As mentioned in the previous review, the construction of variation-aware genome graphs (line 206 ff) seems to neglect recommendations on variant prioritization (<https://doi.org/10.1186/s13059-018-1595-x>). The ancestry of the 294 samples is unclear. Do they indeed represent "global breed diversity"? Referring to Dutta et al. is not enough, as the ancestry of the samples is critical for variant prioritization to enable constructing informative graphs.

Are the 11 million variants detected in the 294 cattle also representative for the samples from Angus, NDama, and Sahiwal that were used for mapping and variant calling evaluations?

We had actually made the ancestry of the 294 samples clear via the form of two PCAs in Figure 1. These illustrate they cover the main Indicine to Taurine clines (PC1) and African Taurine to European Taurine clines (PC2). We have now added a further reference to this figure at line 212. Regarding whether the 11M variants are representative of the samples analysed, we determined the overlap between the 11M variants called across these 294 samples using GATK with the variants identified in the nine samples using VG5p. 2.25M (Angus32065) to 5.12M (Sahiwal_6) of these 11 million variants were found using VG5p. These numbers are highly comparable to the total numbers of

variants found in animals of the same breeds in the original GATK calls. This suggests the 11 million variants cover the overwhelming majority of variants found in these nine animals despite not being in the original set of 294.

Table 1 - Number of variants, out of the 11M carrying the alternative alleles in the reference (i.e. the 294 individuals used to construct the graph) when called using GATK (ref samples) and the samples called using VG (test samples). Each breed is represented by a different colour: Angus (yellow lines), N'Dama (blue lines) and Sahiwal (grey lines)

Ref/Test	Breed	Sample ID	N variants with alternative allele
Ref	Angus	Angus-186	2,416,452
	Angus	Angus-261	2,400,157
	Angus	Angus-294	2,432,483
	Angus	Angus-342	2,334,176
	N'Dama	NDama-ND064	2,495,341
	N'Dama	NDama-ND118	2,758,904
	N'Dama	NDama-ND131	2,357,184
	N'Dama	NDama-ND158	2,467,691
	N'Dama	NDama-ND166	2,570,302
	N'Dama	NDama-ND183	2,806,125
	N'Dama	NDama-ND719	2,760,631
	N'Dama	NDama-SRS1705842	2,539,604
	N'Dama	Ndama_SN010414	3,257,966
N'Dama	Ndama_SN012031	2,706,482	
Sahiwal	Sahiwal-Sha3b	5,346,450	
Test	Angus	Angus01	2,944,613
	Angus	Angus32065	2,253,878
	Angus	Angus34122	2,264,088
	N'Dama	NDama_ND21	2,494,990
	N'Dama	NDama_ND23	2,550,821
	N'Dama	NDama_ND39	2,933,693
	Sahiwal	Sahiwal_3	5,063,082
	Sahiwal	Sahiwal_6	5,121,843
Sahiwal	Sahiwal_7	5,017,298	

Breed-wise, we identify 3,409,133, 3,827,821 and 6,602,728 variants for the Angus, N'Dama and Sahiwal, respectively. When all the 9 samples are combined together (combined through bcftools merge -m all), we have a total of 7.5M variants out of the 11M that are present. Consequently a large amount of variation in these animals is captured by these 11 million variants. Considering that the 11M are found as polymorphic in several breeds, we believe that the 11M variants have been selected appropriately, especially in the light of a multi-breed graph genome.

Previous studies (in different species including cattle, <https://doi.org/10.1186/s13059-020-02105-0>) showed that adding random variants to a genome graph does not lead to any improvements over linear mapping. Talenti et al. report that VG1p does not lead to mapping improvements over VG1, and this is likely because the variants were not prioritised.

Unless we are misinterpreting it we believe the paper the reviewer refers to actually shows that a vg graph with all variants included (i.e. the highest allele frequency threshold of 0% in Figure 2D)

has a lower mapping error than when no variants are included using either vg or BWA. Around a 0.075% mapping error rate with all variants as compared to ~0.09% with none. The authors make this point themselves “The proportion of reads with mapping errors decreased significantly with the number of variants added to the genome graph (Fig. 2d, Pearson R = 0.94, P < 10⁻¹⁶.” The reviewer is correct that some level of variant prioritisation does seem to further reduce mapping errors, but the change is arguably relatively modest, decreasing mapping errors from ~0.075% with all variants included, to ~0.072% when variants are prioritised i.e. a change of 0.003% (also see Figure SN1 in paper referred to above). This modest improvement needs to be balanced against the fact that the prioritisation needs to be done specifically for each project and the breeds and samples being studied. This not only increases the complexity of the analysis but also could potentially lead to problems when comparing results between studies, i.e. if they have been analysed with respect to different reference graphs. It is also worth mentioning that the 11M variants included in the VG1p graph had a minor allele frequency among all the samples of 5%, being in effect an allele-base prioritization for our dataset, which is the point of minimum mapping error shown in Figure 2D of Crysanto and Pausch (2020).

However, we have also added an analysis of the impact of variant prioritisation (see lines 214-219 and 708-711) and a section to the discussion (see lines 444-450). Consistent with the above, we only saw a very modest impact of variant prioritisation. Also, we note we have made publically available all the data required for users to prioritise variants to include in our genome graph if desired. Even though we agree with the reviewer that a breed-specific graph genome may prove more efficient for a single breed study, it would need the generation of multiple graphs for each population considered. This proves to be an especially relevant constraints if we consider the computational burden needed to generate the graph itself prior to the alignments. Therefore, in this study we aimed at generating a series of graphs that the scientific community, in particular those in LMICs or working on LMIC breeds, can take advantage of immediately, regardless of the population.

In addition, a haplotype index may further improve mapping and resolve mapping ambiguity (<https://doi.org/10.1093/bioinformatics/btz575>).

We thank the reviewer for the suggestion. We had actually raised this with the developers of the tools at the beginning of our analyses, but they hinted that the index was not necessary (see github issue <https://github.com/vqteam/sv-genotyping-paper/issues/6>). Considering this, we decided to keep the generation as consistent as possible removing the GBWT index from all the analyses.

Talenti et al. state in the discussion that graph genomes do not lead to significant improvements in small variant calling (l. 428). This is a contradiction to what is presented in their results particularly to Figure 4b,c. Improvements are likely even higher when variants are prioritized appropriately.

We don't believe this statement is inconsistent with Figs 4B and C in that the lines for VG5 (no small variants added) and VG5p (small variants added) in 4C are largely completely overlapping and the heights of the bars in the VG5 and VG5p panels Fig 4B are highly similar. However, we have now expanded the discussion at lines 445-451. See above regarding variant prioritisation. We agree with the reviewer that prioritizing variants would potentially improve the mapping, but it would sacrifice the “generalist graph” approach we were aiming for.

Did ARS-UCD1.2+ also increase mapping ambiguity?

The use of the ARS-UCD1.2+ does increase the number of secondary mappings (see table below).

SAMPLE	PRIMARY (ARS-UCD1.2+)	PRIMARY (ARS-UCD1.2)	PRIMARY INCREASE	SECONDARY (ARS-UCD1.2+)	SECONDARY (ARS-UCD1.2)	SECONDARY INCREASE
HF3457 (BCELL)	469,621,291	457,627,237	3%	341,085,347	329,189,659	3%
HF3457 (NUCFREE)	1,180,022,346	1,170,695,333	1%	914,603,650	905,593,862	1%
ND230 (BCELL)	658,824,369	642,870,385	2%	507,206,668	491,414,738	3%

NELORE1 (BCELL)	940,165,780	888,566,058	5%	585,717,512	534,480,685	9%
NELORE2 (BCELL)	1,922,158,212	1,861,955,997	3%	1,429,440,702	1,369,696,190	4%
NELORE3 (BCELL)	919,523,927	851,870,958	7%	561,450,852	494,112,339	12%

However, a similar point was addressed in the previous revision, and the current version of the manuscript also considered the uniquely mapped reads, which still provided an increase in the number of mapped reads when compared to ARS-UCD1.2 (lines 397-399, 831-835).

Authors indicate in their revised version that Angus samples from public databases present a lower Ti/Tv ratio (line 271). What does this mean/imply? Is there a problem with these samples? Or is the Ti/Tv ratio low, because the graph does not permit accurate variant detection in taurine samples? Can this be tested with other samples from taurine ancestry?

Although the Ti/Tv ratio is lower in these Europeans samples, than the other samples studied, they are in fact comparable to ratios seen in the graph genome paper by Crysanto et al the reviewer refers to above, where they observed a Ti/Tv ratio of 1.91 among variants called in other European taurine animals in non-repetitive, high confidence non-reference sequence. This is comparable to the ~1.9-2 observed in our study. This doesn't appear to necessarily reflect an issue calling variants across taurine samples as suggested by the reviewer in that the N'Dama samples are also taurine.

Is the "54%" in line 228 (apologies for missing this in my first review) correct? An improvement of 54% in perfectly mapping reads seems exceedingly high.

Many thanks for spotting this. We apologise for this mistake. The proportion are reported in column R in supplementary table 6, and display an improvement of perfectly aligned reads in the range 10-27%. We edited lines 233-234 as appropriate, thank you for spotting this.

Reviewer #3 (Remarks to the Author):

The manuscript has been extensively revised, and my comments have been addressed.

I am satisfied with the responses to my comments on structural variation calling/line specific reference genomes and pleased to see a comparison with Delly is included. However, the arguments given in the rebuttal do not come across properly in the manuscript.

From the rebuttal I understand that VG can only detect SV present in the graph and thus no 'novel' SV from an independent re-sequenced individual, correct? This would explain the large difference in detected SV by both methods. Delly is known to be outputting a large amount of SV with varying confidence to let the end user decide on filtering criteria, so yes many should be filtered and cannot be confirmed by other methods. In the manuscript the focus is given to the specificity of VG which is then not really fair. It is good to highlight that SV you find with VG are more likely to be true SVs as most are confirmed by Delly and OM (which makes sense as they have already been observed in individuals used to construct the graph), however, that it cannot exploit split read and discordant reads to detect novel SV.

So I agree with the argumentation in the rebuttal where this is explicitly indicated, but I like to see those arguments better reflected in the manuscript. Rather than claiming VG is a good SV caller due to its higher specificity compared to Delly (implicitly suggesting Delly is a poor SV caller) it can be nuanced.

We thank the reviewer for pointing this out. We agree that comparing the two methods is not trivial and was the main reason behind the exclusion of these analyses in the first version of the paper. It was not our intention to imply that Delly is a poor SV caller, but just to highlight the advantage of a pangenome, graph-based approach over a linear reference-based approach. We added two

sentences to highlight the fact that Delly allows to call more SVs than VG, and that more SVs overlaps with OM than VG (lines 359-361, 468-470). See more details below.

Below are a few sections in the manuscript that need reformulation with respect to the above mentioned comment.

Line 350-355 & Line 433-435 You are focussing a lot on the specificity here which makes VG the better SV caller, however, you ignore the fact that Delly detected a much higher number of deletions confirmed by OM (3,186 vs 1,887). Delly is known to be very sensitive and outputting as many SV as possible to leave the filtering to the user for its specific purpose.

We thank the reviewer for pointing this out. We added the following sentence at line 359-361 to reflect the this:

“This higher number of SVs called by Delly is probably reflective of its ability to take advantage of split reads and unaligned reads to identify novel SVs in the sample.”

We also added a second sentence at lines 468-470 as follow:

“It should be noted that Delly was able to call a large number of potential SVs not present in the graph. Further augmenting the graph with more genomes should though reduce this number.”

These complement lines already in the previous version of the manuscript such as at lines 352-356:

“It should be noted though that Delly called 2,219 SVs overlapping an SV in the OM data not identified by vg. These are potentially sample-specific SVs, that being absent from the graph will be largely uncalled by vg. Further improvements to the graph, for example by including further assemblies, would be expected to reduce this number.”

Line 356-360 This doesn't feel like a fair example when indicating that Delly didn't detect it. You use an insertion as example here. Insertions were filtered for Delly because they had imprecise breakpoints (line 331-333).

We thank the reviewer for pointing this issue. The insertion was not detected by Delly even considering the raw data in any of the samples considered in the study, which is why we kept it as an example of the potential of graph genomes over standard SV callers. We edited the sentence at lines 369-370 to reflect this:

“This SV was identified by both OM samples (Figure 5A), the three re-sequenced N'Dama genomes (Figure 5B) and was present as an alternate sequence in the graph but not identified by Delly, even without filtering any SVs from the different samples”

Figure 4D: I don't understand the numbers given at the intersection, shouldn't they be equal for the methods included in the intersection?

We understand the reviewer's concern about the number. It is often the case that one structural variant in one method overlaps two or more SVs in another, making a direct comparison complicated. We have clarified this in the figure legend (lines 1076-1080).

Textual comment: L603 change 'We the applied...' to 'We then applied...'
L652 I believe '<60bp' should be '>60bp'

We changed as suggested (lines 634 and 682), thank you for spotting this.

REVIEWERS' COMMENTS

Reviewer #1 (Remarks to the Author):

The authors have addressed the majority of my remaining concerns in this revision. I have only a two minor, compulsory points that must be addressed in a subsequent revision:

Line 445: Please cite the "previous studies" mentioned in the text.

Line 452: This sentence contains confusing phrasing and awkward terminology. I recommend that it be revised to paraphrase the concept that I believe the authors intended: that the problem of detection of SNP and INDEL variants from sequence alignment to linear reference genomes has been adequately addressed by existing algorithms, and that the true improvements offered by graph-based references are with the detection of larger structural variants.

Reviewer #2 (Remarks to the Author):

Andrea Talenti and colleagues present a modified version of the manuscript "A cattle graph genome incorporating global breed diversity" that was prepared taking into account comments from three reviewers.

The authors now also report results from a graph that was constructed from a subset of prioritised variants. They also discuss limitations of population-specific graphs, and clarify the aims of their contribution.

During two rounds of revisions, the authors constructively considered the comments / concerns raised by reviewers, which I believe considerably improved the quality and clarity of the manuscript.

In conclusion, I don't have any further comments. The paper reports on a valuable resource for the cattle genomics community. I congratulate Andrea Talenti and colleagues for their great contribution!

Reviewer #3 (Remarks to the Author):

I have no further comments.

RESPONSE TO REVIEWERS' COMMENTS:

Reviewer #1 (Remarks to the Author):

The authors have addressed the majority of my remaining concerns in this revision. I have only a two minor, compulsory points that must be addressed in a subsequent revision:

Line 445: Please cite the "previous studies" mentioned in the text.

Thank you, we added the missing references at the end of the sentence: "*Previous studies have shown that the prioritization of variants included in the graph can potentially lead to further improvements in alignment accuracy^{17,31}.*" (now line 442).

Line 452: This sentence contains confusing phrasing and awkward terminology. I recommend that it be revised to paraphrase the concept that I believe the authors intended: that the problem of detection of SNP and INDEL variants from sequence alignment to linear reference genomes has been adequately addressed by existing algorithms, and that the true improvements offered by graph-based references are with the detection of larger structural variants.

We have now substantially rephrased the sentence to clarify this at line 448: "*While variants calling of SNPs and small InDels is appropriately addressed by standard algorithms, graph genomics improves both read mapping and calling of larger structural variants thanks to the presence of non-reference sequences across the different assemblies.*"